



# Myrmekite and strain weakening in granitoid mylonites

Alberto Ceccato[1]*, Luca Menegon[2], Giorgio Pennacchioni[1], Luiz Fernando Grafulha Morales[3]

[1] Department of Geosciences, University of Padova, 35131 Padova, Italy
[2] School of Geography, Earth and Environmental Sciences, University of Plymouth, PL48AA Plymouth, UK
[3] Scientific Centre for Optical and Electron Microscopy (ScopeM) - ETH Zürich, Switzerland

*Correspondence to*: Alberto Ceccato (alberto.ceccato.2@phd.unipd.it)

* Now at: School of Geography, Earth and Environmental Sciences, University of Plymouth, PL48AA Plymouth, UK

**Abstract.** At mid-crustal conditions, deformation of feldspar is mainly accomplished by a combination of fracturing, dissolution/precipitation and reaction-weakening mechanisms. In particular, K-feldspar
is reaction-weakened by formation of strain-induced myrmekite - a fine-grained symplectite of plagioclase and quartz. Here we investigate with EBSD the microstructure of a granodiorite mylonite, developed at 420-460 °C during cooling of the Rieserferner pluton (Eastern Alps), to assess the microstructural processes and the role of weakening associated with myrmekite development. Our analysis shows that the crystallographic orientation of the plagioclase of pristine myrmekite was
controlled by that of the replaced K-feldspar. Myrmekite nucleation resulted in both grain size reduction and ordered phase mixing by heterogeneous nucleation of quartz and plagioclase. The fine grain size of sheared myrmekite promoted grain size-sensitive creep mechanisms including fluid-assisted grain boundary sliding in plagioclase, coupled with heterogeneous nucleation of quartz within creep cavitation pores. Flow laws calculated for monomineralic quartz, feldspar, and quartz +
plagioclase aggregates (sheared myrmekite), show that during mylonitization at 450 °C grain-size-sensitive creep in sheared myrmekite accommodated strain rates several orders of magnitude higher than monomineralic quartz layers deforming by dislocation creep. Therefore, diffusion creep and grain size-sensitive processes contributed significantly to bulk rock weakening during mylonitization. Our results have implications for modelling the rheology of the mid-upper continental (felsic) crust.

## 1. Introduction

Localization of ductile strain within rocks arises from weakening associated with grain size refinement processes by dynamic recrystallization, metamorphic reactions, and microfracturing (e.g. de Bresser et al., 2001 and reference therein). Grain size reduction, accompanied by phase mixing in polymineralic rocks at high strains, commonly results in a switch of deformation mechanism from
grain-size-insensitive (GSI) to grain-size-sensitive (GSS) creep – one of the most effective strain weakening mechanisms within shear zones (Kruse and Stünitz, 1999; Kilian et al., 2011; Menegon et



al., 2013). Feldspars locally form the load-bearing framework of continental crust rocks (Handy, 1994). At mid-crustal conditions, feldspar deformation mainly occurs by microfracturing and dissolution/precipitation processes, typically associated with metamorphic reactions (Behrmann and Mainprice, 1987; Ree al., 2005; Viegas et al., 2016). K-feldspar is commonly replaced by myrmekite

– a fine-grained symplectic aggregate of quartz and plagioclase (Becke, 1908; Vernon, 1991). Myrmekite replacement is related either to K-feldspar chemical instability in the presence of metasomatic fluids (Cesare et al., 2002), or triggered by intracrystalline strain in K-feldspar during deformation (Simpson and Wintsch, 1989; Menegon et al., 2006). This replacement is acknowledged as a weakening mechanism during ductile deformation of granitoid rocks (Simpson and Wintsch, 1989;

Tsurumi et al., 2003; Menegon et al., 2006; De Toni et al., 2016) and is particularly remarkable when it affects coarse-grained pegmatite (LaTour and Barnett, 1987; Pennacchioni, 2005; Pennacchioni and Zucchi, 2013). Deformation and shearing of myrmekite result into a fine-grained plagioclase + quartz aggregate, that is manifestly weaker than original coarse K-feldspar (Tsurumi et al. 2003; Ree et al., 2005; Ciancaleoni and Marquer, 2006; Viegas et al. 2016). Fine grain size and the local occurrence of

fluid promote phase mixing processes and the development of ultramylonites (Vernon, 1991; Kilian et al., 2011, Czaplińska et al., 2015). The recognition of the key role of myrmekite in strain localization has not been accompanied with a quantitative analysis of the deformation mechanisms within myrmekite-derived fine-grained plagioclase + quartz aggregates. Here we present the detailed analysis of myrmekite evolution from breakdown of K-feldspar into plagioclase and quartz, grain size reduction

and strain localization in the mylonites of the Rieserferner granitoid pluton (Eastern Alps). In this pluton, ductile shear zones nucleated along joints that were locally filled with quartz- and epidote-veins, during post-magmatic cooling (Ceccato et al., 2017; Ceccato and Pennacchioni, 2018). The progressive development of granodiorite mylonite included consumption of K-feldspar by myrmekite leading to increasingly interconnected, fine-grained plagioclase + quartz layers. Selected

microstructures of granodiorite mylonite have been analysed to characterize: (i) the process of myrmekite nucleation; (ii) the deformation mechanisms during myrmekite shearing and transition to plagioclase + quartz aggregates; (iii) the deformation mechanisms of pure-quartz layers; (v) the deformation mechanisms of K-feldspar porphyroclasts and of K-feldspar neoblasts during mylonitization. Furthermore, the application of mixed flow-laws of the aforementioned deformation

mechanisms for polymineralic aggregates allowed the quantification of the extent of rheological weakening resulting from the deformation of myrmekite.



## 2. Geological setting and field description

The tonalitic-granodioritic Rieserferner pluton (Eastern Alps) (Bellieni, 1978) emplaced at ~15 km depth (0.4 GPa; Cesare et al., 2010) into the Austroalpine nappe system at 32 Ma (Romer and Siegesmund, 2003). During post-magmatic cooling, a main set of ductile shear zones developed

exploiting precursor shallowly ESE-dipping joints and the locally associated quartz and epidote veins filling the joints (Ceccato, 2018; Ceccato and Pennacchioni, 2018). Nucleation of ductile shear zones on precursor tabular layers (e.g. dykes, veins) or "surface" discontinuities (joints) has been observed in many granitoid plutons (Adamello: Pennacchioni, 2005; Sierra Nevada: Segall and Simpson, 1986; Pennacchioni and Zucchi, 2013) and in meta-granitoid units (Pennacchioni and Mancktelow, 2007,

2018, and reference therein). The temperature of ductile shearing in the Rieserferner has been estimated at 420-460 °C based on thermodynamic modelling (Ceccato, 2018). Where associated with precursor joints filled with epidote, ductile shearing resulted in cm-thick heterogeneous shear zones with a sigmoidal-shaped foliation in the host granodiorite (Ceccato and Pennacchioni, 2018).

## 3.  Sample description and microstructure

Polished thin sections of granodiorite mylonite were prepared, from rock chips cut parallel to the lineation and perpendicular to the shear plane (XZ plane of finite strain ellipsoid), for the study of the microstructure and of crystallographic preferred orientations (CPO). The microstructural and CPO analysis conducted via EBSD were complemented with microchemical analyses performed with an electron microprobe. Mineral abbreviations after Kretz (1983). Description of methods and analytical

conditions are reported in Appendix A.

The Rieserferner granodiorite consists of quartz, plagioclase, K-feldspar, biotite, allanite/epidote, hornblende, apatite and titanite. The magmatic plagioclase displays oscillatory zoning with a range in composition between $An_{58}$ (core) to $An_{32}$ (rim), and is arranged in glomeroclasts, included in K-feldspar ($Or_{93} - Ab_7$). Various grain-size reduction mechanisms accompanied the development of a

mylonitic foliation in the granodiorite: (i) recrystallization of quartz and biotite (Fig. 1a,b); (ii) formation of myrmekite after K-Feldspar (Fig. 1c,d) and (iii) microfracturing of feldspar and (iv) formation of plagioclase ($An_{29}Ab_{71}Or_{<1}$) – titanite – muscovite symplectite at biotite-plagioclase boundaries (Pennacchioni et al., 2006; Johnson et al., 2008). Pristine myrmekite makes transition to fine grained aggregates of dominant plagioclase + quartz extended into the foliation (Fig. 1b,e). The

mylonitic foliation is defined by alternating layers of: (i) monomineralic quartz; (ii) plagioclase ($An_{26}Ab_{74}Or_{<1}$), quartz, K-feldspar; and (iii) biotite – recrystallized biotite/plagioclase (Fig. 1a).



Syn-kinematic K-feldspar neoblasts ($Or_{96} - Ab_4$) are found in strain shadows around porphyroclasts or dilatant fractures, and are in turn locally replaced by myrmekite (Fig. 1d). With increasing strain, there is a decrease in volume percentage of K-feldspar from 19 vol% (undeformed rock and protomylonite), to 1-6 vol% (mylonite and ultramylonite) (Figs. 2). As counterbalance, there is an increase in volume percentage of fine-grained myrmekite and derived plagioclase + quartz aggregates, from 3% (undeformed rock and protomylonite) to as much as 13% (mylonite and ultramylonite) (Figs. 2b). Ultramylonites consist of a fine-grained (ca. 10 µm grain size) matrix of quartz, plagioclase, biotite, epidote, K-feldspar, titanite, apatite ± garnet ± white mica.

## 4. EBSD and cathodoluminescence analysis

EBSD maps and crystallographic orientation data are reported in Figs. 3 and 4. Results of phase spatial distribution analysis are reported in Fig. 5. Results of image analysis of grain size and shape are reported in Figs. 6 and 7.

### 4.1 Pristine myrmekite

Characteristics of pristine myrmekite are: (i) the preferential development along grain boundaries parallel to the mylonitic foliation, despite locally mantling entirely the K-feldspar porphyroclast (Fig. 1b-e); (ii) the lobate shape protruding into the K-feldspar (Fig. 1c); (iii) the monocrystalline structure of plagioclase within each lobe (20 to 50 µm in size: Fig. 7a), embedding vermicular quartz (sections up to 3 µm in equivalent size: Fig. 6a); (iv) the rather constant spacing between the quartz vermicules of about 3-5 µm across the entire lobe; (v) the preferential elongation of the quartz vermicules orthogonal to the myrmekite/K-feldspar boundary.

The EBSD analysis of the crystallographic relationships between the K-feldspar and the replacing myrmekitic plagioclase and quartz shows that: (i) K-feldspar and plagioclase commonly show similar crystallographic orientations ($(100)_{Kfs} \parallel [100]_{Plg}$, $(010)_{Kfs} \parallel (010)_{Plg}$, $[001]_{Kfs} \parallel (001)_{Plg}$; Fig. 3b, c); (ii) the quartz vermicules do not share any crystallographic plane or direction with K-feldspar or myrmekitic plagioclase (Fig. 3b-c-d); (iii) quartz vermicules within a myrmekite lobe usually have a crystallographic preferred orientation (Fig. 3d) (Abart et al., 2014); and (iv) Dauphiné and Albite twins are occasionally observed in quartz and plagioclase, respectively.

The plagioclase of myrmekite lobes exhibits rare low angle boundaries (misorientations >2°, >5°) that abut against the quartz vermicules (Figs. 3a and 4a). However, the internal distortion of myrmekitic plagioclase is very small (<1°; Fig. SOM1a).





### 4.2 Sheared myrmekite: plagioclase + quartz aggregates

Shearing of myrmekite gave rise to plagioclase + quartz aggregates (± rare K-feldspar and biotite) elongated into the foliation (Fig. 1e). These aggregates are referred hereafter as sheared myrmekite.

Quartz grains of sheared myrmekite occur either as isolated single grains at triple/quadruple junctions
of plagioclase grains or, less commonly, as polycrystalline aggregates elongated normal to the foliation (Fig. 4a). The quartz grain size is around 3 µm (Area B in Fig. 4a; Fig. 6b), but locally increases to >10 µm (Area C in Fig. 4a; Fig. 6c). Individual grains show polygonal, equant shapes (1.5<AR<1.75) or a weak shape preferred orientation (SPO) oriented at low angle to the local mylonitic foliation (Fig. 6e). Quartz grains within sheared myrmekite have no CPO (Fig. 4b), show little internal distortion and
rarely show low angle boundaries with scattered misorientation axis distribution (Fig. 4c). Misorientation angle distribution for correlated pairs displays higher frequency for misorientation < 15° and at 60° than a random-pair distribution (Fig. 4d). The uncorrelated misorientation angle distribution approaches the random-pair distribution.

Plagioclase grains (average grain size of about 7 µm: Figs. 7b-c) are mainly polygonal and range in
shape from almost equant to elongated (1.75<AR<2). Elongated grains define an SPO almost parallel to the local mylonitic foliation (Fig.7d for Area B in Fig. 4a). Plagioclase grains do not show any obvious CPO (Fig. 4e), they display little internal distortion and rare low angle boundaries. The low and high angle misorientation axes are almost uniformly distributed in crystal directions (Fig. 4f). Even though very close to random-pair distribution, correlated misorientation distribution exhibits two weak
peaks at very low angles (<5-10°) and close to 180° (Fig. 4g). Misorientations <70° occurs with slightly higher frequency than the random-pair distribution. Albite-twins, and related 180° misorientations are rarely observed inside new grains (Figs. 3a-4a). In cathodoluminescence (CL) both myrmekitic plagioclase and quartz have a CL grey shade similar to the surrounding non-myrmekitic plagioclase and quartz (similar to Hopson and Ramseyer, 1990) (Fig. 1f,h).

### 4.3 K-feldspar aggregates in strain shadows

K-feldspar neoblasts occur in strain shadows around feldspar porphyroclasts, as well as dispersed within the sheared myrmekite (Figs. 3a, SOM2a; Area E in Fig. 4a). In strain shadows, the orientation of (100), (010), [001] planes of the neoblasts is similar to that of the porphyroclast (Fig. SOM2). The grain size of K-feldspar dispersed within sheared myrmekite is ca. 7 µm, comparable to that of the
plagioclase in the surrounding sheared myrmekite (Fig. SOM2a-f). The analysed K-feldspar aggregate shows a CPO for (010) planes close to the Y kinematic axis (Fig. SOM2d), which is similar to the



orientation of (010) in the adjacent porphyroclast. Misorientation axis/angle distributions show very few scattered data without any clear clustering (Fig. SOM2e).

The CL imaging of K-feldspar grains and porphyroclasts highlights a complex microstructure, in which porphyroclasts exhibit a homogeneous bright shade overprinted by a complex low-grey CL
pattern (Fig. SOM3). K-feldspar grains in sheared myrmekite and tails around porphyroclasts show a homogeneous low-grey CL signature (Fig. SOM3). K-feldspar aggregates elongated parallel to the foliation and enveloped by sheared myrmekite are characterized by bright irregularly-shaped K-feldspar cores (porphyroclasts) surrounded by low-grey shade K-feldspar.

### 4.4 Quartz layers along foliation

Monomineralic quartz layers defining the mylonitic foliation (Figs. 3, 4 and SOM4) show a variable grain size and a shape preferred orientation (SPO), inclined to the foliation consistent with the sense of shear. Dauphiné twin boundaries are widespread (red boundaries in Fig. SOM4a). The quartz c-axis CPO defines an asymmetric Type-I girdle roughly normal to the local mylonitic foliation (Fig. SOM4b). The pole figures of c-axis and <a> directions show maxima close to the Y and to the X
kinematic directions, respectively. Misorientation axis distribution for low angle misorientation (<10°) exhibits a wide maximum close to c-axis and <π−π'> directions in crystal coordinates. They are preferentially oriented slightly off-set from the Y-kinematic direction in sample coordinates (Fig. SOM4c). High angle misorientation axis distributions do not show any clear systematic pattern, except for misorientations around 60°. Misorientation angle distribution (Fig. SOM4d) shows two peaks at
very low angle misorientations (<10°) and around 60° for correlated misorientations. Un-correlated misorientation angle distribution is close to the random-pair distribution.

Quartz layers along the foliation show a variable grain size, usually ranging between 10 µm and 120 µm, mimicking a bimodal grain size distribution with maxima centred respectively at 20-35 µm and 50-70 µm (Figs. 6f and SOM5). The coarser grain sizes (>40 µm) is observed close to the centre of
quartz layers. These grains are usually characterized by subgrains ranging is size between 20 and 35 µm. The smaller grain size (<40 µm) commonly envelope the coarser grains, in addition to prevail at the boundary between monomineralic quartz layers and sheared myrmekite, or around feldspar porphyroclasts (Figs. 3, 4 and SOM5e). CPOs and misorientation data of coarser grains do not differ from those of finer grains. In CL images the quartz layers display an overall homogeneous signature,
with lower-grey shades close to inclusions and layer boundaries (Fig. SOM3).





## 5. Image Analyses

The results of image analysis of EBSD phase maps indicate that pristine and sheared myrmekite have the same phase ratio with ca. 18 vol% of quartz. We have analysed the phase spatial distribution of plagioclase and quartz in both pristine and sheared myrmekite to define their deviation from random

distribution, either towards a clustered or an anticlustered distribution (Heilbronner and Barrett, 2014). Phase spatial distribution in deformed bimodal aggregates in shear zones is interpreted to reflect the activity of specific deformation mechanisms (Kruse and Stünitz, 1999; Menegon et al., 2013). Dislocation creep usually results in either monomineralic aggregates or clustered phase distributions (Heilbronner and Barrett, 2014). Diffusion creep in polymineralic aggregates is commonly

accompanied by heterogeneous phase nucleation that promotes phase mixing and a high degree of anticlustering in phase distribution (Kilian et al., 2011; Menegon et al., 2013).Phase spatial distribution analysis of a two-phase aggregate compares the cumulative lengths of phase boundaries (boundaries between grains of a different phase) and of grain boundaries (boundaries between grains of the same phase) with those expected for a random distribution of the two phases. We have considered three

types of boundaries: (i) plagioclase – plagioclase grain boundaries; (ii) quartz – quartz grain boundaries; and (iii) plagioclase – quartz phase boundaries. The results show that (Fig. 5), in pristine and sheared myrmekite: (i) the surface area fraction of quartz ranges between 0.55-0.75 and 0.55-0.65, respectively; (ii) quartz–quartz grain boundaries occur with a probability lower than for a random distribution indicative of an anticlustered distribution; (iii) plagioclase–plagioclase grain boundaries

occur with a higher probability than for a random distribution indicative of a more clustered distribution; (iv) plagioclase + quartz aggregates display an ordered/anticlustered distribution, with plagioclase – quartz phase boundaries occurring with higher probability than for random distribution of phases.

## 6. Discussion

### 6.1 Formation and shearing of myrmekite

### 6.1.1 Crystallographic relationship between K-feldspar and myrmekitic phases

The EBSD analysis indicates that the K-feldspar and the overgrowing myrmekitic plagioclase have a similar crystallographic orientation, though with some scattering (Fig. 3b; Wirth and Voll, 1987). This suggests the occurrence of a topotactic replacive process where $(100)_{Kfs} \parallel [100]_{Plg}$, $(010)_{Kfs} \parallel (010)_{Plg}$,

$[001]_{Kfs} \parallel (001)_{Plg}$. The scatter in crystallographic orientation between K-feldspar and myrmekitic plagioclase is interpreted to have resulted from deformation during and after myrmekite formation (see



section 6.1.2). The crystallographic orientation of myrmekitic plagioclase and quartz was not controlled by neighbour plagioclase or quartz grains previously in contact with the K-feldspar, differently from what reported by other authors (Stel and Breedveld, 1990; Abart et al., 2014). As observed by Abart et al. (2014), the different myrmekite quartz vermicules have a similar

crystallographic orientation. The anticlustered phase spatial distribution of pristine myrmekite is related to the process of heterogeneous phase nucleation during myrmekite formation (Wirth and Voll, 1987).

### 6.1.2. Transition from pristine- to sheared myrmekite (plagioclase + quartz aggregates)

The transition from pristine to sheared myrmekite was a dynamic process and here we try to constrain

the processes involved as inferred from microstructural changes. These microstructural changes include: (i) randomization of plagioclase CPOs observed in pristine myrmekite; (ii) refinement of plagioclase grain size distribution from scattered in pristine (3 to 50 µm) to homogeneous and centred at 7 µm in sheared myrmekite (Figs. 3, 4 and 7a); (iii) coarsening of quartz grains from <3 µm vermicules to as large as 10 µm rounded-to-polygonal grains in sheared myrmekite (Fig. 6). These

processes are probably related to the minimization of interfacial energy in the vermicular microstructure of the pristine myrmekite (e.g. Odashima et al., 2007; Dégi et al., 2010). Quartz grain coarsening occurred as a consequence of annealing of the pristine vermicular microstructure after the reaction front moved further into the K-feldspar (Fig. 3a). Quartz coarsening implies simultaneous grain size refinement of plagioclase, which probably involved both annealing and micro-fracturing, as

suggested by misorientation analysis on the few low and high misorientation angle boundaries and CPO randomization (Figs. 3, SOM1a). Microfractures could have originated from stress concentrations within the 3-D geometrically/mechanically composite structure of myrmekite (see figure 2 of Hopson and Ramseyer, 1990; Dell'Angelo and Tullis, 1996; Xiao et al., 2002). Annealed myrmekite are then sheared along the mylonitic foliation from the contractional sites around the K-

feldspar porphyroclast.

### 6.2. Deformation mechanisms in the Rieserferner mylonites

### 6.2.1. Sheared myrmekite

Plagioclase and quartz of sheared myrmekite both display: (i) a weak CPO; (ii) rare low angle

boundaries without systematic pattern of misorientation axis distribution; (iii) correlated and un-correlated misorientation angle distributions close to the theoretical random-pair distribution. All these features suggest very limited deformation by dislocation creep in both minerals (Kruse et al., 2001;




Miranda et al., 2016). In addition, the sheared myrmekite show: (i) fine-grained plagioclase and quartz with polygonal, equant to slightly elongated shape (AR<2); (ii) aligned grain boundaries (over the scale of several grain diameters) and common triple/quadruple-junctions; (iii) anticlustered spatial distribution of plagioclase and quartz. These microstructural features are consistent with GSS creep,

including fluid-assisted grain boundary sliding (GBS) (White, 1977; Stünitz and Fitz Gerald, 1993; Fliervoet et al., 1997; Jiang et al., 2000; Wheeler et al., 2001; Lapworth et al., 2002; Bestmann and Prior, 2003; Kilian et al., 2011; Menegon et al., 2013). The occurrence of quartz in triple-quadruple junctions and quartz aggregates elongated orthogonal to the foliation suggest the activity of creep cavitation and heterogeneous quartz nucleation during GSS creep of plagioclase (Fusseis et al., 2009;

Herwegh et al., 2011; Kilian et al., 2011). Heterogeneous phase nucleation in creep cavities led to the anticlustered phase spatial distribution observed in sheared myrmekite (Fig. 5) (Hiraga et al., 2013; Menegon et al., 2015). The constant plagioclase grain size of sheared myrmekite may then result from the combination of initial spacing between quartz vermicules in pristine myrmekite, diffusion creep processes and second-phase grain-boundary pinning during shearing (Herwegh et al., 2011). GSS

processes, phase mixing and second-phase grain-boundary pinnng inhibit grain growth and stabilizes grain size, hindering the efficiency of dynamic recrystallization processes and self-sustaining the activity of GSS processes.

### 6.2.2. K-feldspar tails and neoblasts

K-feldspar is abundant in the low-strain portions of the analysed mylonite (Fig. 2). K-feldspar

porphyroclasts and tails do not show any microstructure, CPO or misorientation axis distribution referable to dislocation creep processes (Fig. SOM2c; Menegon et al., 2008, and reference therein). The similar crystallographic orientation between feldspar(s) porphyroclasts and either K-feldspar tails or fine neoblast aggregates can be explained invoking epitaxial nucleation and growth during dissolution – precipitation (Fig. SOM2). Dissolution – precipitation would be consistent with the K-

feldspar aggregate microstructure observed under CL, which probably reflect either the different chemistry, or the different intragranular strain, observed between magmatic ($Or_{93}Ab_7$) and synkinematic K-feldspar ($Or_{96}Ab_4$) (Ramseyer et al., 1992; Götze et al., 1999; Słaby et al., 2008). The modification of the inherited CPO in fine-grained aggregates could be then related to the occurrence of anisotropic dissolution – precipitation processes and GBS/rigid body rotation during myrmekite

shearing (Behrmann and Mainprice, 1987; Menegon et al., 2008, 2013).





### 6.2.3. Monomineralic quartz layers

The microstructures, CL signatures and strong crystallographic preferred orientation of monomineralic quartz layers indicate deformation by dominant dislocation creep aided by subgrain rotation (SGR) recrystallization (e.g. Fliervoet et al., 1997; Wheeler et al., 2001; Stipp et al., 2002; Bestmann and

Pennacchioni, 2015). The misorientation axes distributions suggest the preferential activation of {m}<a> and {r-z}<a> slip systems (e.g. Ceccato et al., 2017 and references therein).

The analysis of the grain orientation spread (GOS), useful to distinguish different generation of relict and/or recrystallized grains (Cross et al., 2017), suggests that there are no meaningful correlations between grain size and average grain distortion. This missing correlation may reflect a non-steady-

state quartz microstructure during a prolonged deformation history or, more likely, the development of the microstructures at different temperature conditions during pluton cooling. The bimodal grain size of recrystallized quartz includes coarser grains that we infer developed during the relatively high-temperature bulk solid-state deformation of the host granodiorite predating the development of localized shear zones at 450 °C dominated by SGR recrystallization (Ceccato et al., 2017; Ceccato and

Pennacchioni, 2018). Coarser grains in quartz layers (grain sizes from >40) record differential stresses < 40 MPa and strain rates of $10^{-13} - 10^{-14}$ s$^{-1}$. Subgrain and finer grains (20-35 µm in diameter) suggest that localized deformation and shearing occurred at differential stresses close to 40-70 MPa and strain rates of $10^{-11} - 10^{-12}$ s$^{-1}$ (Stipp and Tullis, 2003; Cross et al., 2017).

### 6.3. The rheology of the Rieserferner mylonites

The rheological effect of transformation of coarse K-feldspar to fine-grained sheared myrmekite and the transition to an interconnected, weak, fine grained microstructure (Handy, 1990) is estimated here by investigating the deformational behaviour of different mixtures of plagioclase and quartz, deforming via different contribution of dislocation creep and diffusion creep. Our simplified model does not include biotite, white mica and biotite + plagioclase aggregates.

The bulk strain rate ($\dot{\varepsilon}_{bulk}$) of a mineral aggregate is given by:

(1)    $\dot{\varepsilon}_{bulk} = \dot{\varepsilon}_{Disl} + \dot{\varepsilon}_{Diff}$

where: $\dot{\varepsilon}_{Disl}$ and $\dot{\varepsilon}_{Diff}$ represents the strain rates of dislocation creep and diffusion creep of mineral components, respectively. The flow law of Hirth et al. (2001) has been used to calculate the contribution of dislocation creep of quartz:



(2) $\qquad \dot{\varepsilon}_{q-Disl} = A_q f_h \sigma^n e^{\left(-\frac{Q_q}{RT}\right)}$

where: $A_q$ is the pre-exponential factor for quartz (MPa$^{-n}$ s$^{-1}$); $f_h$ is the water fugacity coefficient; $\sigma$ is the differential stress (MPa); $n$ is the stress exponent; $Q_q$ is the activation energy (J); $R$ is the gas constant (J/K*mol); $T$ is the temperature (K). Following Platt (2015), the flow law of den Brok (1998) for thin-film model of pressure-solution has been used to calculate the contribution of pressure-solution to quartz deformation:

(3) $\qquad \dot{\varepsilon}_{q-Diff} = \dot{\varepsilon}_{qps} = C_2 \frac{\rho_f}{\rho_s} \frac{\sigma}{d^3} \frac{V c D_w}{RT}$

where: $C_2$ is a shape constant; $\rho_f$ and $\rho_s$ are the fluid and solid densities (Kg m$^{-3}$), respectively; $d$ is the grain size (µm); $V$ is the molar volume (µm$^3$ mol$^{-1}$); $c$ is the solubility of the solid in the fluid phase (molar fraction); $D_w$ is the diffusivity of the solid in the grain-boundary fluid film (µm$^2$ s$^{-1}$). The flow laws of Rybacki et al. (2006) have been used to calculate the contribution of dislocation and diffusion creep of feldspar:

(4) $\qquad \dot{\varepsilon} = A_f f_h \frac{\sigma^n}{d^m} e^{\left(-\frac{Q_f + pV^{act}}{RT}\right)}$

(5) $\qquad \dot{\varepsilon}_{f-Disl} = A_f f_h \sigma^3 e^{\left(-\frac{Q_f + pV^{act}}{RT}\right)}$

(6) $\qquad \dot{\varepsilon}_{f-Diff} = A_f f_h \frac{\sigma}{d^3} e^{\left(-\frac{Q_f + pV^{act}}{RT}\right)}$

where: $A_f$ is the pre-exponential factor for feldspar (MPa$^{-n}$ µm$^m$ s$^{-1}$); $d$ is the grain size (µm); $m$ is the grain-size exponent ($m=3$ for diffusion creep; $m=0$ for dislocation creep); $p$ is the confining pressure (MPa); $V^{act}$ is the activation volume (m$^3$ mol$^{-1}$). Deformation mechanisms maps for sheared myrmekite were calculated following the self-consistent approach presented in Dimanov and Dresen (2005) and Platt (2015), in which flow law parameters for the poly-phase aggregate deforming via dislocation creep are recalculated as follows:

(7) $\qquad \mathrm{Log}_{10}\, n_a = \phi_1 \log_{10} n_1 + \phi_2 \log_{10} n_2$

(8) $\qquad Q_a = [Q_2(n_a - n_1) - Q_1(n_a - n_2)]/(n_2 - n_1)$

(9) $\qquad \mathrm{Log}_{10}\, A_a = [\log_{10} A_2(n_a - n_1) - \log_{10} A_1(n_a - n_2)]/(n_2 - n_1)$




where: $n_a$ is the stress exponent for dislocation creep of the two-phase mixture; $\phi_i$ is the volume fraction of the phase i; $n_i$ is the stress exponent of the $i$ phase; $Q_a$ is the dislocation creep activation energy for the aggregate (J); $Q_i$ is the activation energy for the $i$ phase (J); $A_a$ is the pre-exponential factor for the aggregate (MPa$^{-n}$ µm$^m$ s$^{-1}$); $A_i$ is the pre-exponential factor for the $i$ phase (MPa$^{-n}$ µm$^m$ s$^{-1}$). In order to

account for water fugacity and activation volumes parameters in feldspar flow laws (Rybacki and Dresen, 2006), we implemented the model of Platt (2015), which includes the activation volume for feldspars in the calculation of the aggregate activation energy $Q_a$, as follow:

(10)    $Q_a = [(Q_f + pV^{act})(n_a - n_Q) - Q_Q(n_a - n_f)] \, / \, (n_f\text{-}n_Q)$

where: $p$ is the confining pressure (MPa); $V^{act}$ is the activation volume for feldspars (m$^3$/mol). Water

fugacity coefficients were integrated in the resulting flow law of sheared myrmekite deforming by dislocation creep:

(11)    $\dot{\varepsilon}_{Disl} = A_a f_h \sigma^{na} e^{(-\frac{Q_a}{RT})}$

The flow law for sheared myrmekite deforming by diffusion creep (plagioclase) and thin-film pressure solution (quartz) has been calculated, following the approach of Dimanov and Dresen (2005) and Platt

(2015), by considering pressure-solution (in quartz) and diffusion creep (in feldspar) to contribute linearly to the bulk viscosity of the aggregate, $\mu_a$:

(12)    $3\mu_a{}^2 + [2(\mu_q+\mu_f) - 5(\phi_q\mu_q + \phi_f\mu_f)]\mu_a - 2\mu_q\mu_f = 0$

where

(13)    $\mu_q = \dfrac{\sigma}{2\dot{\varepsilon}_{qps}}$

is the viscosity of quartz deforming via pressure-solution processes calculated following the thin-film model of den Brok (1998); and

(14)    $\mu_f = \dfrac{\sigma}{2\dot{\varepsilon}_{f-diff}}$

 is the viscosity of feldspar deforming via diffusion creep. Therefore, the deformation mechanism maps of the sheared myrmekite have been calculated using the following flow law:

(15)    $\dot{\varepsilon}_{bulk} = A_a f_h \sigma^{n_a} e^{(-\frac{Q_a}{RT})} + \dfrac{\sigma}{2\mu_a}$



The rheological modelling of poly-phase aggregates containing more than two-rheological phases (e.g. a granitoid composed of plagioclase + quartz + myrmekite) has been performed applying iteratively the calculation of Platt (2015). For example, for a mixture composed of $\phi_X$, $\phi_Y$ and $\phi_Z$ volume fractions of the phases X (plagioclase), Y (quartz), and Z (sheared myrmekite):

(i)     Firstly, $n_{XY}$, $Q_{XY}$, $A_{XY}$ flow law parameters for the XY two-phase mixture are calculated following equations (7), (8) and (9), adopting the volume fractions $\phi_{X1}$ and $\phi_{Y1}$ defined as follow:

(16)     $\phi_{X1} = \phi_X / (1 - \phi_Z)$; $\phi_{Y1} = \phi_Y / (1 - \phi_Z)$.

(ii)     Then for the calculation of the three-phase mixture flow law parameters $n_{XYZ}$, $Q_{XYZ}$, $A_{XYZ}$,

considering the three-phase mixture as the result of mixing between phases "XY" and Z, volume fractions are recalculated as follow:

(17)     $\phi_{XY} = \phi_X + \phi_Y$; $\phi_Z = \phi_Z$.

and the parameters are calculated as follow:

(18)     $n_{XYZ} = 10^{\wedge}(\phi_{XY} \log_{10} n_{XY} + \phi_Z \log_{10} n_Z)$

(19)     $Q_{XYZ} = [Q_Z(n_{XYZ} - n_{XY}) - Q_{XY}(n_{XYZ} - n_Z)]/(n_Z - n_{XY})$

(20)     $A_{XYZ} = 10^{\wedge}[\log_{10}A_Z(n_{XYZ} - n_{XY}) - \log_{10}A_{XY}(n_{XYZ} - n_Z)]/(n_Z - n_{XY})$.

For the calculation of the rheology of an aggregate in which dislocation and diffusion creep contribute in different proportions to the total strain rate (granitoid rock including a variable amount of myrmekite, see Discussion below), a limiting factor $\theta$ has been introduced in equation (15). For

example, for an aggregate in which diffusion creep is limited to a specific volume proportion of phases:

(21)     $\dot{\varepsilon}_{bulk} = A_a f_h \sigma^{n_a} e^{(-\frac{Q_a}{RT})} + \theta_{diff} \frac{\sigma}{2\mu_a}$

where $\theta_{diff}$ represents the volume fraction of the phases undergoing diffusion creep in the aggregate. To consider the progressive transformation of feldspar into myrmekite with increasing strain, differential stress vs. strain rate curves have been calculated for a "granitoid" aggregate with increasing

vol% of myrmekite substituting for feldspar (Fig. 8c). The progression of the reaction is quantified by the reaction progress factor $\chi$. The maximum volume percentage of feldspar substitution has been limited to

(22)     $\phi_{MAX} = 20$ vol% ;

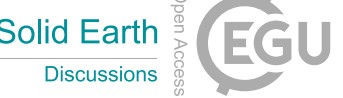


the average concentration of K-feldspar in granite and granodiorite. For $\chi=0$ (no myrmekite), the rock is composed by 60 vol% plagioclase $An_{100}$ and 40 vol% quartz. For $\chi>0$, plagioclase $An_{100}$ (representing K-feldspar) is increasingly replaced by myrmekite. Myrmekite, plagioclase and quartz volume proportion in the rock are then calculated respectively as follow:

(23)    $\phi_{Myrm}{}^1 = \chi * \phi_{MAX}$;

(24)    $\phi_{Plg}{}^1 = \phi_{Plg} - \chi * \phi_{MAX}$;

(25)    $\phi_{Qtz}{}^1 = \phi_{Qtz}$

The calculations represent the case in which grain size sensitive creep occurs only in sheared myrmekite, whereas granitoid quartz and feldspars deform only by dislocation creep. Therefore, the

contribution of diffusion creep to bulk viscosity in equation (21) is proportional to

(26)    $\theta_{diff} = \chi * \phi_{MAX}$

Deformation mechanisms maps have been calculated and plotted on grain-size vs. differential stress and on differential-stress vs. strain rate diagrams for the following three compositions (Fig. 8):

(i)      monomineralic quartz layer deforming via both dislocation and diffusion creep (Fig. 8a);

(ii)      sheared myrmekite, modelled as 80 vol% plagioclase ($An_{60}$) + 20 vol% quartz deforming via both dislocation creep and grain size sensitive creep (Fig. 8b); the input grain size is 7 μm, identical for both minerals;

(iii)     a mixture of 60% plagioclase ($An_{100}$) + 40% quartz assumed as a simplified composition representative of a mica-free granitoid rock deforming only by dislocation creep (after

20          referred as "granitoid") (Fig.8c);

The above general flow laws and flow law parameters are estimated for the pressure-temperature conditions of mylonitization of the Rieserferner (450°C and 0.35GPa; Ceccato, 2018). At these conditions, the calculated water fugacity is $f_h = 97$ MPa (Pitzer and Sterner, 1994). Fluid density, quartz solubility and diffusivity in the thin-film (grain boundary) fluid has been calculated following Fournier

and Potter (1982) and Burnham et al. (1969). The flow law parameters defined for $An_{100}$ and $An_{60}$ by Rybacki and Dresen (2004) have been adopted for our calculations to simulate different compositions of "granitoid" and myrmekitic feldspars. In our calculations All the K-feldspar has been considered as plagioclase, given the lack of flow law parameters for K-feldspar (see discussion in Platt, 2015; Viegas





et al., 2016). Our calculation includes the contribution of GBS to the bulk strain rate of the feldspar aggregate, which is considered in the flow law parameters adopted here (see discussions in Xiao et al., 2002; Rybacki and Dresen, 2004).

### 6.3.1. Relative strength, rheological ranking and strain localization

The grain-size vs. differential stress and differential stress vs. strain rate diagrams for the three above defined aggregates are shown in Fig. 8. Flow law parameters are listed in Table 1. The results indicate that the three aggregates can be ranked, from the strongest to the weakest, as follows: (i) quartz-feldspar "granitoid" aggregate; (ii) monomineralic quartz aggregates (grain size of 4-10-20-100 µm); (iii) sheared myrmekite. This ranking is consistent and validated by several field and microstructural

observations, that highlight the strain localization capability of monomineralic quartz layers (i.e. quartz veins) and two-phase microstructural domains (i.e. sheared myrmekite) in granitoid rocks (Pennacchioni, 2005; Pennacchioni and Mancktelow, 2007; Menegon and Pennacchioni, 2010; Pennacchioni and Zucchi, 2013; Pennacchioni et al., 2010; Ceccato et al., 2017). The results of rheological calculation of plagioclase + quartz aggregates deforming via diffusion creep (sheared

myrmekite) are consistent and comparable with some of the experimental results of Xiao et al., (2002) extrapolated to geological temperatures (Fig. 8c). The experimental data that best fit our estimated rheological curve are those obtained from triaxial deformation experiments of synthetic very fine-grained wet aggregate of 80 vol% $An_{100}$ (6 µm) + 20 vol% quartz (10 µm). In addition, this rheological ranking supports the interpretation that quartz grains inside pristine myrmekite deforming via

dissolution – precipitation creep may behave as strong inclusions compared to feldspar grains deforming via diffusion creep; around these strong inclusions stress can concentrate and trigger microfracturing in the surrounding plagioclase.

These results show that in the Rieserferner mylonites an effective strength contrast occur between the host rock, mono- and poly-mineralic aggregates occurs as a consequence of the different deformation

mechanisms. To quantify the effective strength contrast between the modelled compositions, we consider both constant stress and constant strain-rate conditions. The differential stress of 40-70 MPa estimated from the finer grain size of quartz (20-35 µm) can be considered representative of the bulk flow stress of the mylonite. At constant stress conditions, such differential stress corresponds to a strain rate of $10^{-11}$-$10^{-13}$ s$^{-1}$ of the quartz aggregates deforming by dislocation creep (Fig. 8a). At such

differential stress, a sheared myrmekite deforming via diffusion creep would flow at strain rates >$10^{-12}$ s$^{-1}$, depending on the actual grain size of the aggregate (red transparent area in Fig. 8b). For the grain size range of sheared myrmekite (4-7 µm), the observed strain rates are always >$10^{-11}$ s$^{-1}$, and for the



above defined differential stress range the calculated strain rate is on the order of $10^{-9}$ s$^{-1}$ (intersection between red transparent area and black box in Fig. 8b). Therefore, assuming constant differential stress conditions, a strain-rate partitioning of 2-4 orders of magnitude is expected between monomineralic quartz and sheared myrmekite (similarly to Behrmann and Mainprice, 1987). Such strain-rate

partitioning at constant stress could also explain the observed decrease in quartz grain size from the core of monomineralic layers toward neighbouring sheared myrmekite (Fig. 4). The differential stress, calculated for sheared myrmekite deforming via diffusion creep at constant strain rate conditions of $10^{-11}$ - $10^{-12}$ s$^{-1}$, is always <45 MPa. Under constant strain rate assumption, the strength contrast between monomineralic quartz and sheared myrmekite is not quantifiable; however, the sheared

myrmekite are always weaker than monomineralic quartz deforming via dislocation creep. Strain rates in the order of $10^{-11}$-$10^{-13}$ s$^{-1}$ would require grain sizes in the range of 10-100 µm in the sheared myrmekite deforming by diffusion creep only (grey shaded areas in Fig. 8b).

### 6.3.2. The effect of myrmekite reaction

Figure 9d shows the different curves describing the rheological behaviour of a simplified granitoid

rock where K-feldspar is progressively replaced, up to 20 vol%, by sheared myrmekite. The flow behaviour of the derived granitoid mylonite is represented by the grey curves, and is linear viscous for most of the investigated conditions. The complete consumption of K-feldspar results in 3-4 orders of magnitude increase of strain rate, consistent with experimental observations (Xiao et al., 2002). A similar increase in strain rate is already observed for reaction progress factor of $\chi = 0.25$, i.e. for a 5

vol% of sheared myrmekite in the total rock volume. These results can be compared to the different degree of myrmekite substitution observed along the strain gradient in the shear zone and also justify the progressive increase in strain toward the ultramylonite with increasing myrmekite substitution (Fig. 2), suggesting positive feedback between strain-induced myrmekite and strain accommodation. Dissolution-precipitation creep of K-feldspar and associated GSS creep in K-feldspar + plagioclase +

quartz aggregates have been already described by Behrmann and Mainprice (1987) as an efficient strain accommodation and weakening mechanism in quartz-feldspar mylonites. In the Rieserferner mylonites, GSS creep of K-feldspar seems to be dominant in protomylonite, but its role decreases with increasing myrmekite substitution (Fig. 2). The positive correlation between accommodated strain and myrmekite substitution suggests that GSS creep processes in K-feldspar are however not capable of

accommodating strain at rates comparable to those produced by GSS creep in sheared myrmekite.

The effective role of small volume fraction of myrmekite in rheological weakening of the Rieserferner granitoid during mylonitization might be overestimated by our calculation, for two main reasons: (i)





other weakening mechanisms may concur during homogeneous deformation of the granitoid rock that are not considered in our simplified model of granitoid (such as feldspar GSS creep, biotite recrystallization); (ii) myrmekite initially occurred into scattered non interconnected pockets (e.g. Handy, 1994); it is only with increasing strain and increasing volume fraction of sheared myrmekite,

that these isolate pockets coalesced to form an interconnected weak layer microstructure. In the Rieserferner mylonites effective interconnection of sheared myrmekite is already developed in presence of 5 to 7 vol% of myrmekite and is particularly well developed at 10-15 vol% (Fig. 2). Therefore, mylonites containing up to 15 vol% of sheared myrmekite ideally underwent deformation at strain rates of $10^{-10}$-$10^{-11}$ s$^{-1}$ and at differential stresses in the range between 14 and 70 MPa. These

mylonites were synkinematic to mylonitic quartz veins described in Ceccato et al. (2017), for which quartz paleopiezometry retrieved comparable strain rates of $10^{-11}$ s$^{-1}$ developed under 117 MPa differential stress (10 µm of grain size).

### 7. Conclusions

Metamorphic reactions contribute importantly to strain weakening within the Rieserferner granitoid

mylonites. A primary grain size reduction mechanism is related to the development of myrmekite evolving, with increasing strain, to weak aggregates of quartz and plagioclase. Topotactic replacement has been inferred from the coincidence between myrmekitic plagioclase and parent K-feldspar grain crystal lattices. Transition from pristine myrmekite to fine-grained sheared myrmekite involved micro-fracturing, annealing and shearing of the resulting granoblastic aggregate. Sheared myrmekite consists

of fine grained plagioclase + quartz aggregates (7 µm and 4 µm in grain size, respectively) that show ordered (anticlustered) spatial distribution and well-defined shape preferred orientation; quartz usually occurs at triple- and quadruple-junction between plagioclase grains. Both plagioclase and quartz show weak CPOs and almost uniform misorientation angle distributions. Sheared myrmekite microstructural features suggest that different deformation mechanisms occur in plagioclase and quartz: plagioclase

deforms mainly by GSS creep, whereas dissolution - precipitation and nucleation processes are dominant in quartz. Myrmekite formation promotes also phase mixing, as the pristine myrmekite microstructure predispose the development of an "anticlustered" spatial distribution of phases in the recrystallized aggregate. Strong grain size reduction and the nucleation of plagioclase + quartz polymineralic aggregates lead to a switch in the dominant deformation mechanisms, activating GSS

creep processes and triggered phase mixing. GSS processes and phase mixing inhibit grain growth and stabilizes grain size, hindering the efficiency of dynamic recrystallization processes and self-sustaining the activity of GSS processes. Therefore, the stress-induced formation of myrmekite lead to the



activation of self-sustaining weakening processes. Results of rheological calculations show that, at the conditions of Rieserferner mylonitization, sheared myrmekite are several orders of magnitude weaker than pure quartz layers and ideal granitoid rock deforming via dislocation creep. Strain-rate partitioning is therefore expected to occur between sheared myrmekite and monomineralic quartz

layers. 5 vol% of myrmekite would lead to an increase in strain rate of 3-4 orders of magnitude. However, the effective role of myrmekite in rock weakening is however dependent on the evolution of the rock microstructure. Effective weakening need for interconnection of sheared myrmekite layers that occur after the development of 10-15 vol% of myrmekite. This work once again highlights the importance of metamorphic reactions and micro-cataclastic processes as grain size reduction

mechanisms in feldspar and their role in localization of ductile deformation. The microstructural results and the rheological calculation presented here will be useful for further development of detailed rheological models of feldspar-rich rocks (continental crust rocks) at mid-crustal conditions.

**Code availability**

The MATLAB script used for rheological calculation is available on request from the first author.

**Data availability**

Supplementary data are available in Supplementary Online Material (SOM).

**Appendix A: Methods**

**A.1 EBSD analysis**

Electron backscattered diffraction analysis was carried out on a JEOL 7001 FEG SEM equipped with

a NordLys Max EBSD detector (AZTec acquisition software, Oxford Instruments) at the Electron Microscope Centre of Plymouth University. The thin section was SYTON-polished for ca. 3 hours and carbon coated. EBSD patterns were acquired on rectangular grids with step sizes of 0.2, 0.3 and 0.35 µm. Working conditions during acquisition of EBSD patterns were 20 kV, 70° sample tilt, high vacuum, and working distance between 17 and 23 mm. All data have been processed and analysed

using CHANNEL5 software of HKL Technology, Oxford Instruments.  Noise reduction was applied following Bestmann and Prior (2003). Local mis-indexing between plagioclase and K-feldspar was resolved by nullifying the subset of selected grains with area $<1\mu m^2$ in each map. Dauphiné twins smaller than 0.5 µm have been interpreted as an error from mis-indexing and were replaced by the average orientation of the neighbouring pixels. The indexed phases and relative symmetry group used



for the indexing are: quartz – Trigonal -3m; plagioclase (anorthite) – Triclinic -1; orthoclase –
Monoclinic 2/m; clinozoisite, biotite and garnet have been indexed where present, but orientation data
have not been analysed. Critical misorientation for the distinction between low- and high-angle
boundaries have been chosen at 10°. Quartz grain boundaries with 60°±5° of misorientation were

disregarded from grain detection procedure, to avoid any contribution from Dauphiné twinning.
Plagioclase grain boundaries with 180°±5° of misorientation around [010] were disregarded from grain
detection procedure, to avoid any contribution from Albite twinning. The pole figures (one-point-per-
grain, where not differently specified) are plotted as equal area, lower hemisphere projections oriented
with the general shear zone kinematics reference system (X = stretching lineation; Z = pole to general

shear plane/vein boundary); whereas the misorientation axis distributions in sample coordinates are
plotted as equal area, upper hemisphere projections. The inverse pole figures for misorientation axis
distribution in crystal coordinates are upper hemisphere projections. Contoured projections have
constant contouring parameters (Halfwidth: 10°). Contouring lines are given only for the 0.5-10 m.u.d.
(multiple of uniform distributions) range.

**A.2 SEM – Cathodoluminescence**

Cathodoluminescence imaging was performed in a FEI Quanta 200 FEI equipped with Gatan monocle
detector. Imaging was performed using an accelerating voltage of 20 kV, beam current of 8 nA and
working distance of 20 mm in C-coated (15 nm) thin sections used for EBSD analysis. To avoid
incorrect interpretation of potential artifacts in the sample, secondary (SE) and backscatter electron

images were collected simultaneously with CL.

**A.3 Grain size analysis**

Grain sizes were obtained from the grain detection routine of the HKL Channel5 Tango software. The
grain size was calculated as diameter of the circle with an equivalent area. The minimum cut-off area
was set to 1 µm$^2$ which means that only grains of a size ≥4 or ≥9 pixels (depending on the map

acquisition step-size) were considered. Grain size data were represented as area-weighted distributions
by plotting frequency against the square-root grain-size-equivalent grain diameters (as in Herwegh and
Berger, 2004; Berger et al., 2011). The grain size distribution approaches a Gaussian distribution when
plotted in this way, allowing a good estimate of the mean grain size. The geometric mean grain size
(red thick line in grain size distribution diagrams) was obtained graphically as the maximum frequency

grain size of the distribution curve. The distribution curve (blue line in grain size distribution diagrams)
was obtained interpolating distribution data with a 6$^{th}$ degree polynomial equation in Excel-MS Office.
Relative frequencies are normalized to 1.

### A.4 Image analysis

Image analysis of grain shape was performed on both SEM-BSE images and phase maps obtained from EBSD. Quantification of phase amount (vol%) was performed through segmentation of SEM-BSE images of a whole thin section collected at the Electron Microscopy Centre of the University of

Plymouth. Image processing and thresholding was done with the ImageJ software, and further processing together with manual correction were applied to improve data quality and to ensure the correspondence of greyscale ranges with specific mineral phases. Grain boundary images and phase distribution images were obtained directly from EBSD phase maps and grain boundary maps elaborated by Channel5 (HKL technology). Before the analysis with ImageJ software, images were

manually corrected in order to exclude mis-indexing and non-indexed orientation pixels. Grain boundaries and phase amount have been quantified by pixel counting.

### A.5 Electron microprobe analysis (EMPA)

Microchemical analyses were performed with EM wavelength-dispersive spectroscopy (WDS) at Electron Microprobe Laboratory at the Università degli Studi di Milano with a Jeol 8200 Super Probe;

the operating conditions were: 15 kV accelerating voltage; 5 nA (feldspar, epidote and phyllosilicate) beam current. PAP correction program was applied to convert X-ray counts into oxide weight percentages.

### Author contributions

LM, GP and AC developed the initial idea of the study and performed initial exploratory SEM study.

GP collected the samples of Rieserferner mylonites. LM acquired EBSD data. AC performed EBSD data processing and analysis, and performed rheological calculations. LFGM performed cathodoluminescence analysis. AC prepared the figures and the manuscript with contributions from all the co-authors.

### Competing interests

The authors declare that they have no conflict of interest.

### Acknowledgements

Simone Papa, Francesco Giuntoli, Luca Pellegrino are thanked for fruitful discussions. Andrea Risplendente is thanked for his help during EMPA data collection at the Università degli Studi di



Milano. The staff at University of Plymouth Electron Microscopy Centre is thanked for the assistance during EBSD data acquisition. Luca Menegon acknowledges the financial support from a FP7 Marie Curie Career Integration Grant (grant agreement PCIG13-GA-2013-618289). Financial support from the University of Padova ("Progetto di Ateneo" CPDA140255) and from the Foundation "Ing. Aldo Gini" is acknowledged.





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





**Figure and Tables Captions**

**Table 1.** Parameters adopted in the rheological calculations. (a) List of the general parameters adopted in the rheological calculations. (b) Values of flow law parameters adopted in the rheological calculations according to mineral phase and deformation mechanism.

**Figure 1.** Microstructures of Rieserferner granodiorite mylonites. (a) Microphotograph (crossed nichols) showing the layered structure of granodiorite mylonites, composed of alternating layers of recrystallized quartz, recrystallized biotite + plagioclase + quartz, and plagioclase + quartz. White arrows indicate layers of recrystallized quartz (upper) and biotite (lower). (b) SEM-BSE image of the area shown in (a). (c) SEM-BSE image of a pristine myrmekite replacing K-feldspar. (d) SEM-BSE
image of the K-feldspar + biotite tails in strain shadows between two plagioclase porphyroclasts. K-feldspar in the strain shadows is in turn replaced by myrmekite (white arrows). (e) SEM-BSE image of a K-feldspar porphyroclast and of sheared myrmekite. Pristine myrmekite developed on K-feldspar boundaries parallel to the mylonitic foliation are sheared to form plagioclase + quartz aggregates (sheared myrmekite). The white polygon encloses K-feldspar neoblasts in strain shadows and sheared
myrmekite. (f) CL image of (e). Note the alteration of the CL signal after the EBSD scan. (g) K-feldspar and sheared myrmekite aggregate (particular of the EBSD map of Fig. 3). (h) CL image of (g).

**Figure 2.** Phase distribution across a strain gradient in a granodiorite mylonite. (a) Mosaic of SEM-BSE images with the K-feldspar and the myrmekite + sheared myrmekite coloured in red and pale
blue, respectively. The yellow rectangles indicate the location of the EBSD maps of Figs. 3, 4 and SOM2. (b) Bar diagram showing the volume percentage of K-feldspar (red bars) and myrmekite (pale blue bars) across the microstructure: PM = protomylonite; M = mylonite; and  UM = ultramylonite.

**Figure 3.** EBSD map and crystallographic orientation data of incipient myrmekite and parent K-feldspar. (a) EBSD-derived phase map. The area delimited by dashed polygons represents pristine
myrmekite. Pole figures for: (b) K-feldspar grains on which pristine myrmekite nucleated; (c) plagioclase and (d) quartz in pristine myrmekite.

**Figure 4.** EBSD map and crystallographic orientation data of pristine and sheared myrmekite of Fig. 1e. (a) EBSD phase map including areas (A, B, C, D) selected for grain size analysis and phase distribution analysis. (b)  Pole figures for quartz from the sheared myrmekite of area B. Upper row:
scattered data. Lower row: contoured data. (c) Misorientation axis distributions for quartz in sample (upper row) and crystal (lower row) coordinate system. (d) Misorientation angle distribution for quartz.



(e) Pole figures for plagioclase from sheared myrmekite of Area B. Upper row: scattered data. Lower row: contoured data. In this case, the [100] plagioclase pole figure is reported in upper hemisphere, where the maximum has been observed. (f) Misorientation axis distributions for plagioclase in sample (upper row) and crystal (lower row) coordinate system. (g) Misorientation angle distribution for
plagioclase.

**Figure 5.** Image analysis on phase spatial distribution. The diagram reports phase- and grain-boundary fraction for sheared myrmekite. Continuous curves represent the theoretical probability of phase- and grain-boundary fraction as a function of quartz content expected for a random distribution in a two-phase aggregate. The small maps on the left hand side report one of the analysed areas (Area C. Fig.
4), showing from the top to the bottom the phase map, the related plagioclase, quartz grain boundaries and phase boundaries.

**Figure 6.** Area-weighted grain size distributions and SPO for quartz. (a) Grain size distribution for quartz in incipient myrmekite A in Fig. 4a. (b) Grain size distribution for quartz in sheared myrmekite B in Fig. 4a. (c) Grain size distribution for quartz in sheared myrmekite C in Fig. 4a. (d) Grain size
distribution for quartz in monomineralic layer in Fig. 4a. (e) Rose diagram showing the orientation of major axis of quartz grains, defining a weak SPO.

**Figure 7.** Area-weighted grain size distributions and SPO for plagioclase. (a) Grain size distribution for plagioclase in myrmekite of Fig. 3. (b) Grain size distribution for plagioclase in incipient myrmekite A in Fig. 4a. (c) Grain size distribution for plagioclase in sheared myrmekite B in Fig. 4a.
(d) Rose diagram showing the orientation of major axis of plagioclase grains, defining a weak SPO.

**Figure 8.** Diagrams obtained from the calculation of the rheological model explained in the text. Grains size vs. Differential stress map with contoured strain rate curves obtained for: (a) quartz, (b) 80% plagioclase (An60) + 20% quartz aggregates. (a) The piezometric curve from Stipp and Tullis (2003) (black curve) and Cross et al. (2017) (red curve) are reported. Red and black stars mark the
differential stress/strain-rate conditions defined by the grain size observed in pure quartz layers: (A) 35 µm; (B) 20 µm; (C) 10 µm (Ceccato et al., 2017). (b) A and B marked red polygons represent the differential stress range obtained from piezometric calculations on pure quartz layers (red and black stars along respective piezometric curves). The black dashed line represents the boundary between dislocation and diffusion creep dominated conditions. The black rectangle represents the grain size
range (4-7 µm) observed in the sheared myrmekite. The grey semi-transparent polygon defines the field of possible grain-size and differential stress conditions for iso-strain-rate conditions defined from piezometric relations. (c) Log Differential stress vs. Log Strain rate diagram reporting the curve



calculated for pure quartz with different grain sizes, sheared myrmekite, ideal granitoid rock and the curves representing the rheology of pure feldspar aggregates. For comparison, one of the curve obtained from experimental data of Xiao et al., (2002) is reported (black dashed curve). Grey field represent the uncertainties on the experimentally defined rheological curve. (d) Log Differential stress
5  vs. Log Strain rate diagram reporting the curve calculated for pure quartz, sheared myrmekite and ideal granitoid rock and the curves representing the rheology of a granitoid (60% An100 + 40% Qtz) with variable amount of sheared myrmekite (80% plagioclase An60 + 20% Qtz). Maximum substitution is limited to 20% of initial feldspar (see text for explanation).





| Parameter | Description | Value | Units | Reference |
|---|---|---|---|---|
| $P$ | Pressure | 350 | MPa | 4 |
| $T$ | Temperature | 723 | K | 4 |
| $f_h$ | Water fugacity | 97 | MPa | 5 |
| $R$ | Gas constant | 8.314 | $J\ K^{-1}\ mol^{-1}$ | |
| $d$ | Grain size | | $\mu m$ | |
| $n$ | Stress exponent | | | 1,2 |
| $m$ | Grain size exponent | | | 1,2 |
| $A$ | Pre-exponential factor | | $MPa^{-n}\ \mu m^{m}\ s^{-1}$ | 1,2 |
| $Q$ | Activation energy | | $J$ | 1,2 |
| $\rho_f$ | Fluid density | 837.5 | $Kg\ m^{-3}$ | |
| $\rho_s$ | Solid density | 2650 | $Kg\ m^{-3}$ | |
| $C_2$ | Shape constnt | 44 | | 3 |
| $V$ | Molar Volume of quartz | 22690000 | $\mu m^{3}\ mol^{-1}$ | |
| $c$ | Solubility of solid in the fluid phase | 0.0014 | molar fraction | 6 |
| $D_w$ | Diffusivity of the solid in the grain-boundary fluid film | 1.604E-11 | $\mu m^{2}\ s^{-1}$ | 3 |
| $V^{act}$ | Activation volume | 0.00038 | $m^{3}\ mol^{-1}$ | 7 |
| $\mu$ | Viscosity | | $MPa\ s^{-1}$ | a |

| Parameter | Dislocation creep | | | Diffusion Creep | | | b |
|---|---|---|---|---|---|---|---|
| Phase | Quartz | Plagioclase | | Quartz | Plagioclase | | |
| Composition | | $An_{100}$ | $An_{60}$ | | $An_{100}$ | $An_{60}$ | |
| n | 4 | 3 | 3 | 1 | 1 | 1 | |
| m | 0 | 0 | 0 | 3 | 3 | 3 | |
| A | 6.30957E-12 | 398.1 | 0.031 | | 50.1 | 12.6 | |
| Q | 135000 | 345000 | 235000 | 137000 | 159000 | 153000 | |
| Reference | 1 | 2 | 2 | 3 | 2 | 2 | |

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

**Table 1**

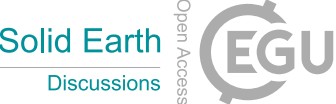



**Figure 1**





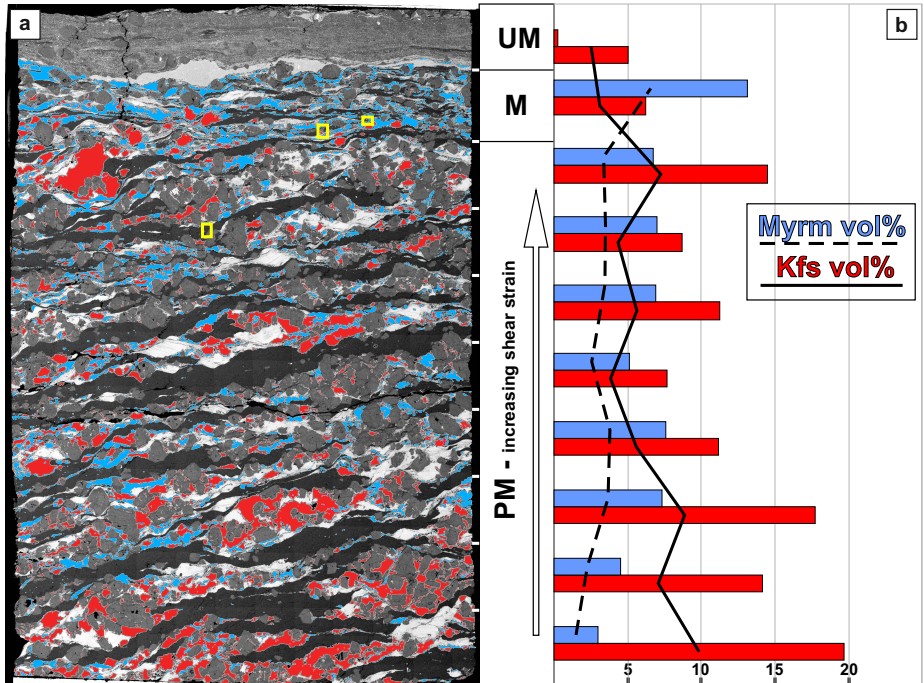

**Figure 2**





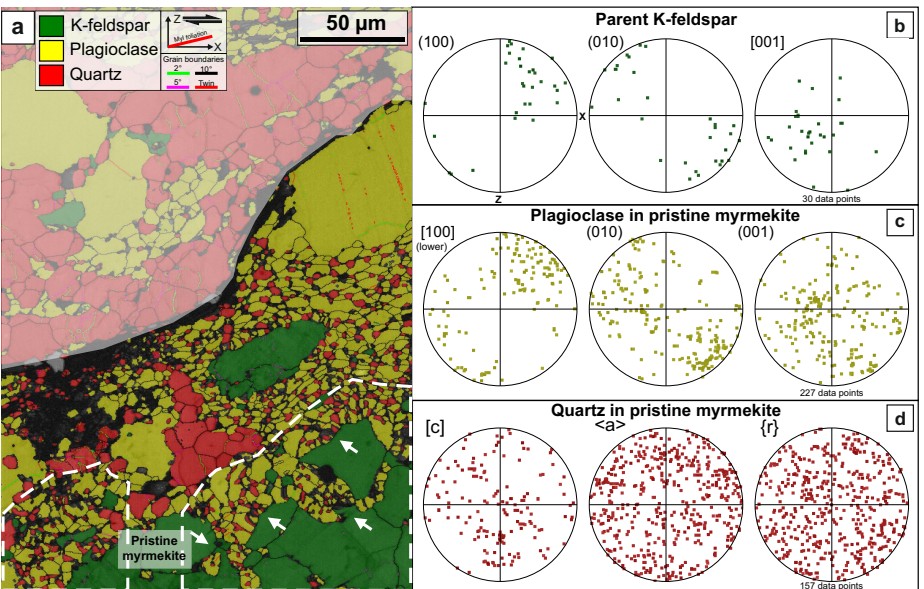

**Figure 3**



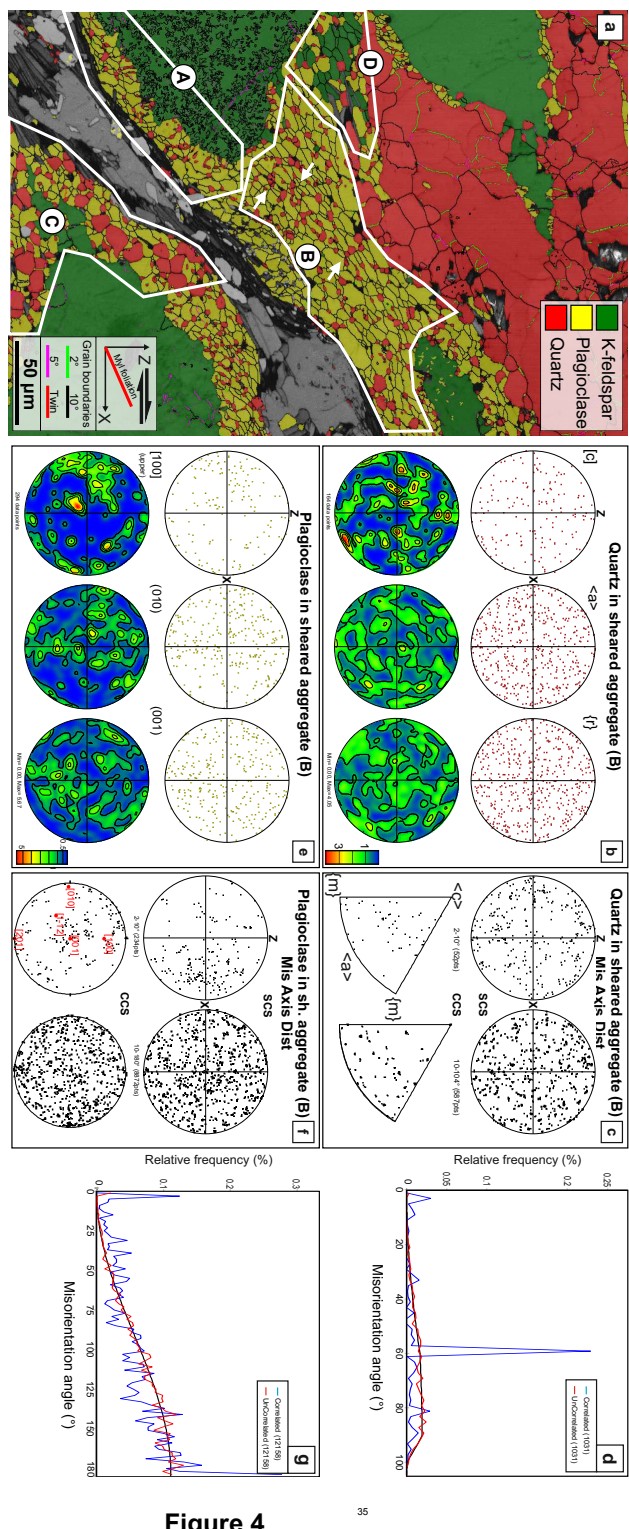

**Figure 4**



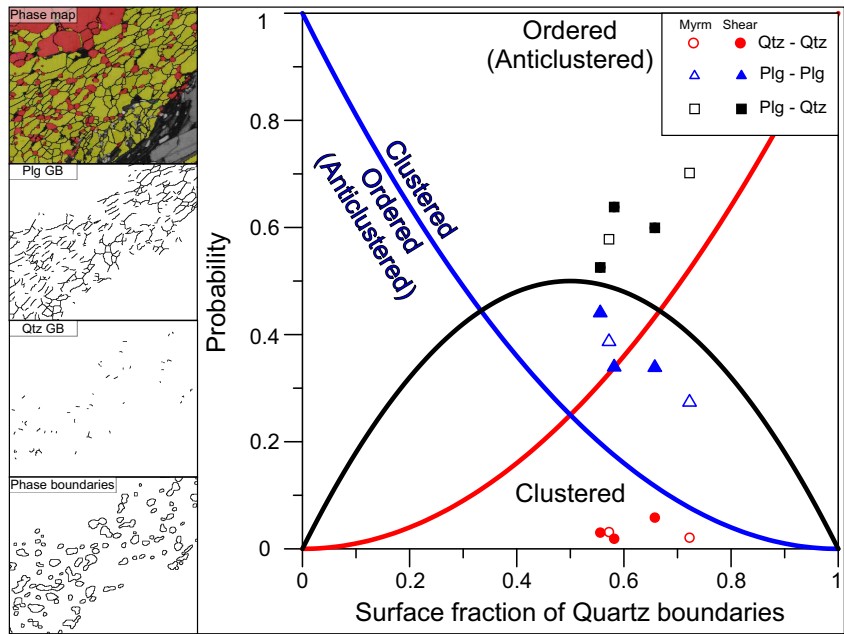

**Figure 5**



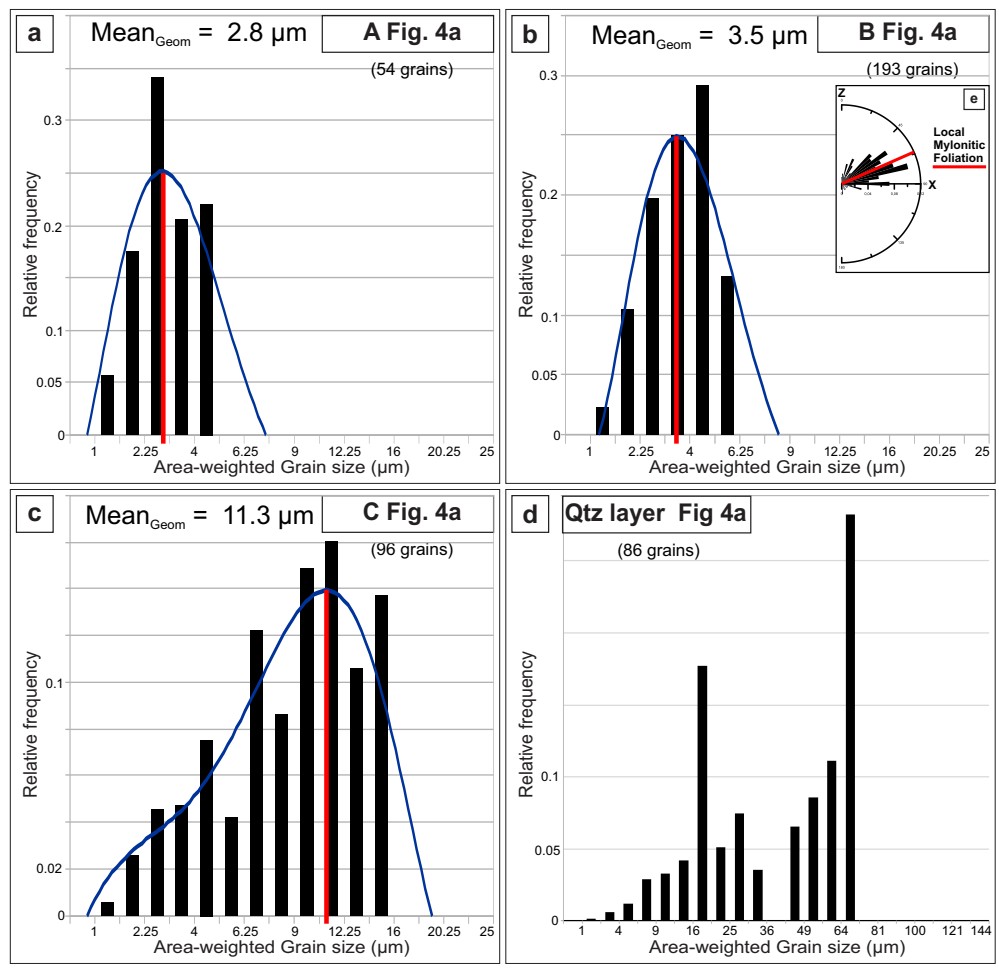

**Figure 6**





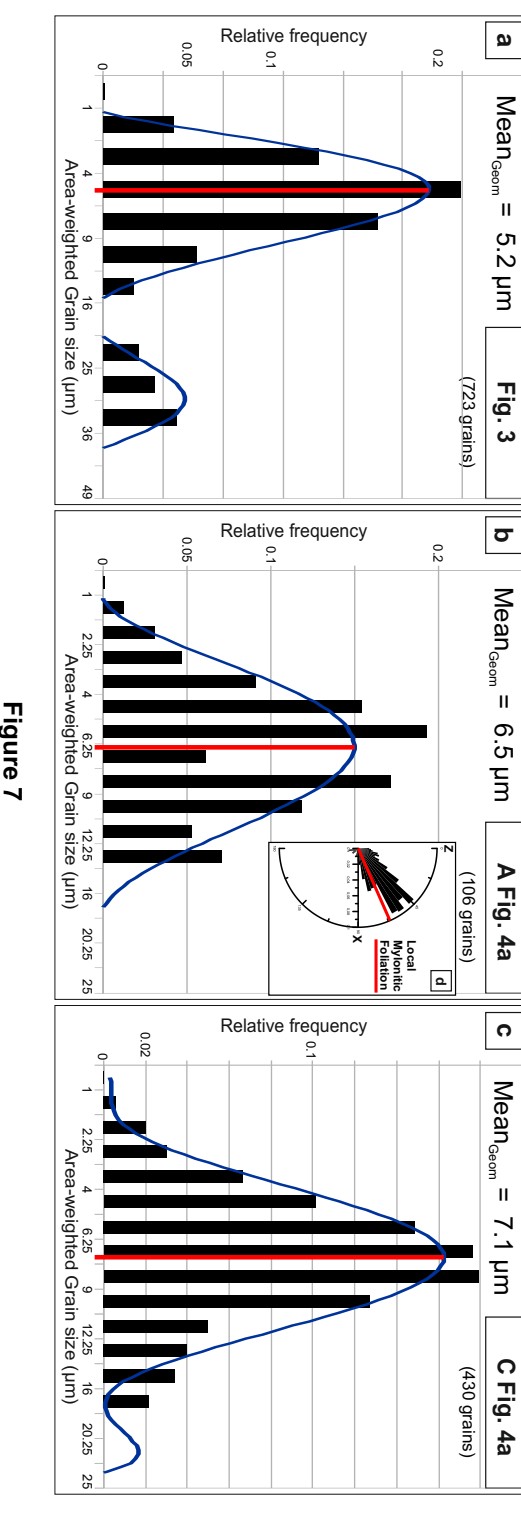

**Figure 7**





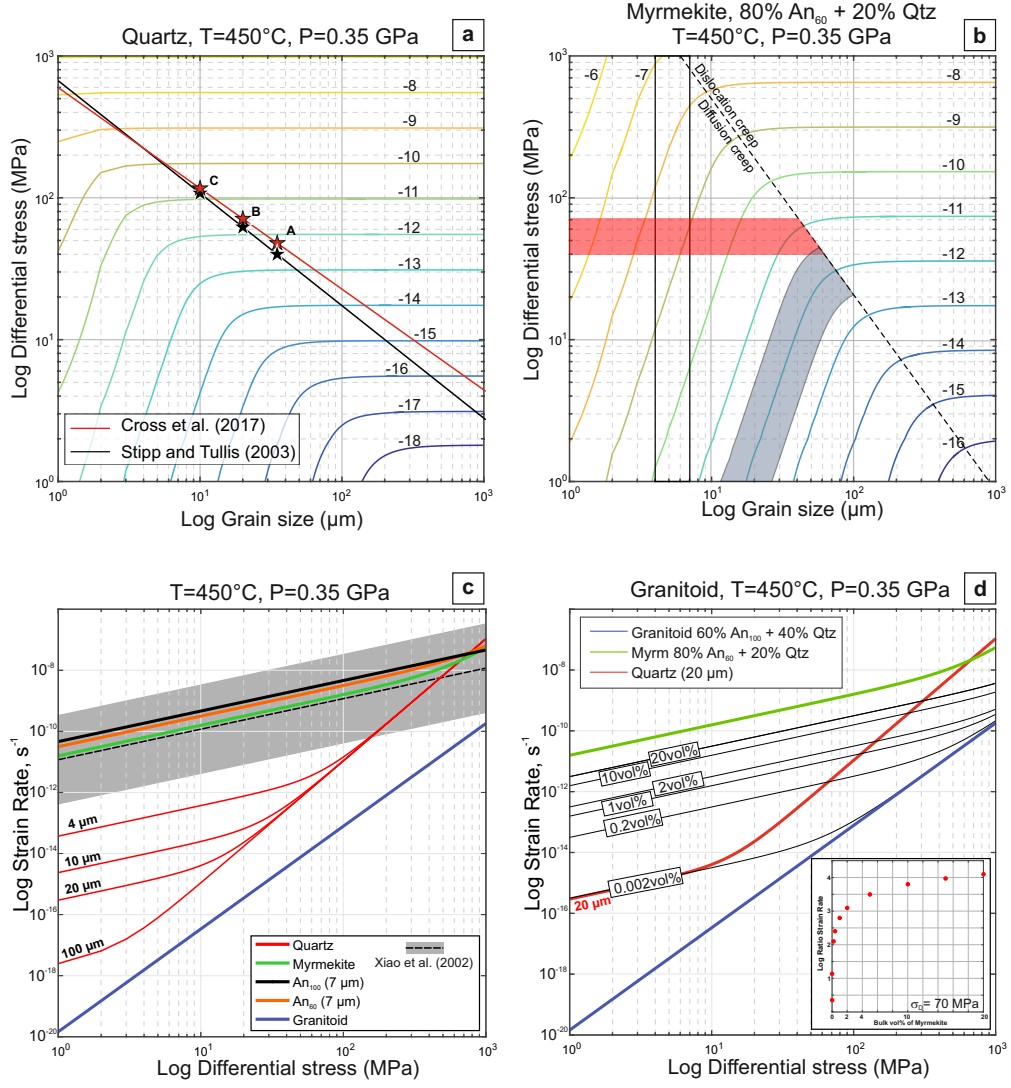

**Figure 8**