# Peer review of "Myrmekite and strain weakening in granitoid mylonites"

_Solid Earth, 2018_

## Referee Comment (RC1) · Anonymous Referee #1 · 2 Aug 2018

General comments This paper described microstructural characteristics of a granodiorite mylonite developed at 420–460 °C during cooling of the Rieserferner pluton (Eastern Alps), and the role of weakening associated with myrmekite development. Based on the results of rheological calculations, the authors found that during mylonitization at 450 °C grain-size-sensitive creep in sheared myrmekite accommodated strain rates several orders of magnitude higher than the model granitoid deforming by dislocation creep, and then contributed significantly to bulk rock weakening during mylonitization. The descriptions of microstructures and textures of feldspars and quartz presented in this paper are robust, and discussion and conclusions are reliable and interesting. This paper contributes to understand deformation process/mechanism of the mid-upper continental (felsic) crust. In this paper, there are so many supplementary fig-

ures. These figures frequently referred in the main text, and then it is complicated and disturbs our understanding the manuscript. Some supplementary figures should be appeared as figures in the manuscript. The order of figures is somewhat strange. The results of image analysis of grain size and shape (Figs. 6 and 7) should be appeared prior to the the results of phase spatial distribution analysis (Fig. 5). Descriptions of the rheological calculations (section 6.3) are little bit complicated, and then they are not easy to understand. I would like the authors to rewrite and reorganize some sentences in the section 6.3. Although the authors described that micro-cataclastic process or micro-fracturing is a dominant grain size reduction mechanism of plagioclase in the samples analyzed here, the microstructural observations indicative of the micro-cataclastic process or micro-fracturing of plagioclase are not described sufficiently.

Specific comments (1) P4, L2–4: In Fig. 2a, there is no identification of myrmekite and K-feldspar for the ultramylonite. Please identify them in Fig. 2a. In Fig. 2b, there are two red bars for the ultramylonite. Is this correct? If so, what do the two different bars represent? (2) P4, L16–17: What is "monocrystalline structure"? This means plagioclase is a single grain? Please clarify the structural characteristics of plagioclase within in each lobe. (3) P4, L25–26: The quartz vermicules do not show any obvious CPO (Fig. 3d). (4) P4, L29: I do not know why "However". Please remove the word. (5) P5, L7: Please define "AR" (6) P5, L15: At least for me, some plagioclase grains in Area B in Fig. 4a is elongated with the aspect ratio of >2. I would like to see the histogram for aspect ratio of plagioclase grains. Related topic also appears in P9, L2. (7) P5, L18; What does "in crystal direction" mean? It means "in the crystal coordinate system"? If so, please rephrase it. (8) P5, L19–20: two weak peaks? two strong (or distinct) peaks! (9) P6, L12–15: If the pole figure of c-axis shows maxima close to Y kinematic direction, the quartz fabric pattern could be assigned to Type-II crossed gridle or single girdle with Y-point maxima. However, the authors described that the quartz fabric pattern was assigned to Type-I crossed girdle (P6, L13). (10) P8, L18–21: In this sentence, it has been described that grain size refinement of plagioclase involves micro-fracturing as suggested by misorientation analysis on the few low and

high misorientation angle boundaries and CPO randomization. However, the authors have not discussed the mechanism of grain size refinement of plagioclase, based on their own microstructural observations. The following paper may be helpful to discuss this issue: Okudaira, T., Shigematsu, N., Harigane, Y. and Yoshida, K. (2017) Journal of Structural Geology, 95, 171–187. (11) P8, L32–P9, L1: Kruse et al. (2001) and Miranda et al. (2016) suggested very limited deformation by dislocation creep for plagioclase aggregates in mylonites. As far as I know, Okudaira and Shigematsu (2012, Journal of Geophysical Research, 117, B03210, doi:10.1029/2011.JB008799) only described very limited deformation by dislocation creep for quartz aggregates in natural mylonites. (12) P9, L20: "... do not show any microstructure" may be "... do not show any deformation microstructure". (13) P10, L9–11: How about the effect of annealing during and or after deformation? The quartz grains associated with myrmekite may be annealed, and then some of quartz grains in monomineralic quartz layer may be also annealed at least partially. (14) P11, L2: fh is water fugacity coefficient, not water fugacity itself? What is water fugacity coefficient? (15) P14, L2: Why would the composition of plagioclase be assumed to be An100, instead of An60? I do not understand the effect of the plagioclase composition. (16) Equations (23), (24) and (25): What does the superscript of 1 mean? Please describe them. (17) P14, L14: In this figure, the result of diffusion creep for quartz is not necessary. (18) P14, L18–20: I cannot understand this sentence and Fig. 8c. This sentence means a mixture of plagioclase and quartz (i.e., ideal granitoid rock) deformed by dislocation creep. The other curves in Fig. 8c are necessary? It is very confusing. Quartz and plagioclase are deformed by diffusion creep? The calculation scheme for myrmekite is similar to those for Fig. 8b? (19) P18, L9: What is the observation indicative of micro-cataclastic process or micro-fracturing as a dominant grain size reduction mechanism of plagioclase in your samples. Please see also my comment (10).

Technical corrections (1) P3, L11: Ceccato (2018) is missed in the reference list. (2) P6, L24: "Figs. 6f and SOM5" should be "Figs. 6d and SOM5".

---

## Short Comment (SC1) · 2 Aug 2018

Although a "metasomatic" role of fluid is accepted or proposed in the origin of myrmekite and similar microstructures, I see the Cesare et al (2002) paper quoted here in the context "Myrmekite replacement is related either to K-feldspar chemical instability in the presence of metasomatic fluids (Cesare et al., 2002),..."

On the contrary, in the conclusions of Cesare et al (2002) one can read: "The agreement of observed mineral compositions and abundances with predictions based on thermodynamic calculations, along with existence of CKNASH mass balance relations among corona and adjacent matrix, supports our contention that myrmekite can develop in a closed system, of thin-section scale, without involving larger scale metaso-

matic exchange."

A revision of the manuscript is necessary. Thanks,

Bernardo Cesare

---

## Author Comment (AC1) · 3 Aug 2018

Dear Bernardo,

Thank you for your interest in the paper and your contribution. The role of fluids and (local?) metasomatism during myrmekite formation processes is somehow "postulated" since the seminal work of Becke (1908) and then also analysed in Phillips (1980). I agree with your comment, the manuscript has been reviewed and modified accordingly to include both point of views. The citation for the metasomatism/myrmekite relationship has been substituted with a more appropriate (Phillips, 1980) for the sake of introducing the subject of the paper.

Thanks,

[Figure]

Interactive

comment

**SED**

Alberto Ceccato

Please also note the supplement to this comment:
https://www.solid-earth-discuss.net/se-2018-70/se-2018-70-AC1-supplement.pdf

**Supplement:**

[revised manuscript text omitted]

---

## Referee Comment (RC2) · E. A. Miranda (Referee) · 17 Sep 2018

General Comments Ceccato et al. use EBSD to characterize the microstructures of naturally deformed granodiorite mylonites from the middle crust, and based on these microstructures, they interpret a progression in activity of the deformation mechanisms that resulted in strain localization. They show that the nucleation of myrmekite resulted in a grain size reduction that ultimately produced phase mixing, leading to grain size sensitive deformation and effective strain localization. This work is an important contribution to our understanding of how quartzofeldspathic rocks deform in the middle crust where the rheology-controlling minerals of the crust (quartz and feldspar) undergo a variety of deformation mechanisms that are aided by metamorphic reaction and reaction with fluids. The microstructural data from the mylonites are of high quality and

described adequately, and they use the deformation conditions of these mylonites to infer rheology from published flow law data. The deformation mechanism maps derived from flow law data allow them to interpret a temporal variation in deformation mechanisms and infer the magnitude of weakening based on differences in strain rates between the lesser-deformed granodiorite and the sheared myrmekite shear zones.

The manuscript would benefit from some substantial reorganization in three main areas: 1) the methods, 2) the figures, and 3) separation of data/interpretation. Regarding the methods: the methods (at least the basics) should be in the main body of the manuscript and not in the supplementary material. As written, there are not any methods in the main body. Some of the post-processing details for EBSD can stay in the supplementary material, but the instruments, working conditions, etc., need to be in the main text. Regarding the figures: there is a tendency for important and primary EBSD data to be relegated to supplementary figures, whereas the flow law derivation in the main body of the manuscript is extremely detailed, and much of it can be moved to the supplementary material for readers who wish to follow it in more detail. I recommend moving the supplementary figure EBSD maps out of the appendix, and incorporating them into the main set of figures within the manuscript. This is important because they are the primary data upon which the interpretations are made. My last point is that there are a number of interpretive statements located in the geologic setting and in the data sections; these should be clearly separated from descriptive data and moved to the discussion.

The writing seems a little rushed, and is not as polished as it could be for the results and interpretations to have maximum impact. Many paragraphs are lacking strong topic sentences, and are instead effectively callouts to the figures. Strengthening the topic sentence would make the results more impactful and easier to understand. In places there are phrases rather than complete sentences, or under-developed paragraphs consisting of one sentence. In addition, there is some awkward or convoluted phrasing, and there are, at times, little continuity of thought between sentences within a

paragraph. The manuscript would benefit substantially from careful revision to address these issues.

I recommend that the authors make these revisions and also consider the more specific comments/recommendations (highlighted below) prior to publication.

Specific Comments: Abstract, Page 1, Line 11. Rephrase for clarity of reading: Here we use EBSD to investigate the microstructure of a granodiorite mylonite...

Page 1, Line 11. Is it important to focus the end of the sentence on weakening of coarse-grained pegmatite? It seems like a distraction given that the focus of the paper is not on pegmatites. It may be best to remove this last phrase so that the sentence is more focused on replacement being a weakening mechanism during ductile deformation, which will provide much more continuity with the next sentence about deformation and shearing of myrmekite.

Page 2, Line 14. There needs to be a stronger link between the sentence that ends with the Viegas et al. reference and the next one that starts with 'Fine grain size'. This should be done with the second sentence (i.e., 'Fine grain size...) so that the weak plag + qtz aggregate is explained as a result of phase mixing and ultramylonites. As written, it does not explicitly link phase mixing and ultramylonites to the samples targeted in this study.

Page 2, Line 16. Rephrase sentence to make it easier to read: Though the key role of myrmekite in strain localization has been recognized, it has not been accompanied with a quantitative analysis, etc...

Page 2, Line 18. The Introduction should be broken up into at least two paragraphs. Make a new paragraph starting with the sentence: 'Here we present the detailed analysis...' so that the objectives of the work are clearly separated from the background and disciplinary context.

Page 2, Lines 18-20. The sentence beginning with 'Here we present...' is awkwardly

written, and needs to do a better job summarizing the findings of the work. As written, I am not sure if the authors wish to emphasize the two-fold nature of the work (e.g., 1) analysis of myrmekite evolution and 2) grain size reduction and strain localization), or if they mean to portray this as the temporal evolution of some process, i.e., starting with myrmekite evolution and finishing with strain localization in the mylonites. The problem hinges on the part of the sentence between the words 'quartz' and 'grain size'. There either needs to be a conjunction word between 'quartz' and 'grain size' (to? and?) because it's not grammatically correct as written. This is why it is difficult to understand whether 'grain size reduction' just applies to quartz, or to both quartz and plagioclase, and calls into question the intent of the authors.

Page 2, Lines 21-23. These sentences are confusing as written, making it hard to understand which mylonites are being studied. If the shear zones nucleate along joints filled with quartz and epidote, where is the K-spar coming from? Are there two types of shear zones, one set of SZs within the granodiorite and the other set in the quartz- and epidote-filled joints? Does one grade laterally into the other? Please revise for clarity.

Page 2, Line 5. I recommend you remove the word 'precursor'. It is implied that the joints are precursory by saying shear zones exploited them.

Page 3, Section 3. Sample description and microstructure. The writing in this section is a little disjointed, with abrupt changes between sentences. Some better linkage between sentences to provide a smoother train of thought would be helpful.

Page 3, Lines 6-9. The sentence beginning with 'Nucleation..' is effectively a comparison of the field descriptions with what has been documented in the literature for other locations. Such a comparative statement is best left to the discussion rather than in a section devoted to field description.

Page 3, Line 13. I think this section is a little incomplete. There is only a statement describing the shear zones for epidote-filled joints, but the authors state in the previous section that there are also quartz-filled joints upon which shear zones nucleate and

also regular joints (no filling) that serve as precursors to shear zones. I recommend that the authors add field descriptions of the shear zones associated with quartz-filled joints and "plain" joints.

Page 3, Lines 15-16. Rearrange the sentence and break into two sentences for clarity: Polished thin sections of granodiorite mylonite were prepared for the study of microstructure and of crystallographic preferred orientation (CPO). The rock chips were cut parallel to the lineation and perpendicular to the shear plane (XZ plane of finite strain ellipsoid).

Page 3, Line 18. Specify which minerals were analyzed by microprobe.

Page 3, Line 20. The description of EBSD and electron microprobe methods and analytical conditions must be reported in the main body of the manuscript, not hidden in the Appendix.

Page 3, Line 22. The sentence beginning with 'The magmatic plagioclase' is awkwardly written, too long, and hard to understand. It reads like the oscillatory zoning has a range in composition rather than the plagioclase having a range in composition.

Page 3, Lines 24-28. The sentence beginning with 'Various grain size reduction...' is phrased in an interpretive rather than descriptive way. Please rephrase in terms of objective description to be consistent with the "Sample description and microstructure" section title.

Page 4, Line 2. There is too abrupt of a change when the data from Figure 2 are introduced. This should be another paragraph, or at least have some more explanatory text about the volume percentage data before it is introduced. Page 4, Line 7-8. Why isn't there a photo of the ultramylonites included in Figure 1? All the rest of the mylonites described in this section have a corresponding picture, so this would be good to include for sake of completeness.

Page 4, Line 10. This section is under-developed. These sentences read like a table

of contents rather than a data section. It should either be fleshed out into paragraphs where the data are explained, or this brief summary of figure content should be merged into the text in Section 4.1 onward.

Page 4, Line 17. Figure 7a is called out before Figures 3, 4, 5 have even been introduced. The figures should either be renumbered so that the call outs are in numerical order, or these out-of-order callouts should be removed. There is another error of this nature in Line 18 with respect to Figure 6a.

Page 5, Line 1. This sentence is strongly interpretive for a data section. It interprets the origin of the sheared myrmekite, but I suggest keeping the sample/microstructure description objective here and to wait for the Discussion to make the interpretation.

Page 5, Line 9. The callout refers to figure 6e, but there is no 6e (only a-d).

Page 5, Line 6. I find the figure callouts hard to follow here. We are flipping between Figure 4 and 6, without 5 being called out yet. Perhaps rearrange the figure order to keep it more organized.

Section 4.3. The paragraphs in this section are lacking strong topic sentences to lead the paragraphs, so that the data are a bit hard to follow. Revise such that a topic sentence gives a summary of the contents of the remainder of the paragraph so that the reader has an idea of what trends are being described.

Page 5, Line 28. Only (100) and (010) are planes. The [001] data are directional and should be described as such.

Page 7, Line 6. "Phase spatial distribution..." This sentence is interpretive and the importance of that interpretation is discussed in the next few sentences. I agree this is important, but the interpretation and its importance in a disciplinary context should be in the discussion rather than in the data section.

Page 10, Lines 15-18. Are these differential stresses calculated from grain size piezometry? How are the strain rates calculated? The delivery of the stress values and strain

rates is not thorough enough here; if grain size piezometry is an analysis taken on in the work, it needs to be explicitly stated.

Page 10, Line 24. There is something grammatically incorrect about this sentence; revise to clarify what is meant by the last phrase.

Page 10, Section 6.3. In the interest of brevity, it might work well to put some of the flow law derivations and calculations into the supplemental online material.

Page 11, Line 11. It is worth noting here that this is a wet plagioclase flow law, and it would be good to quote the amount of water present in these experiments.

Page 15, Lines 5-6. There needs to be a stronger topic sentence than the one that begins Section 6.3.1. As written, it's essentially just a callout to Figure 8, but a more powerful topic sentence would give better direction to the paragraph. The sentence should instead focus on the primary results of the deformation mechanism maps.

Page 15, Line 26. The authors state that they consider both constant stress and constant strain rate conditions, but are there any geologic data from the mylonites that support these assumptions about constant stress/strain rate? This should be justified or explained in a little more detail.

Technical Corrections: Abstract Line14. Plagioclase of in pristine myrmekite.. Intro Line 12. . . .shearing of myrmekite results into in a a fine-grained. . . Intro Line 18. Add comma after 'myrmekite-derived'. Intro Line 29. 'Flow-laws' should not be hyphenated. Intro Line 30. Use present tense (allows) instead of past tense for the verb 'allowed'. Page 3, Line 1. Add the word 'was' before 'emplaced'. Page 3, Line 19. 'Mineral abbreviations after Kretz' is not a complete sentence and should be revised. As written, it reads more like a figure caption than the main body text of a manuscript. Page 3, Line 21. Please make a new paragraph beginning with the sentence 'The Rieserferner granodiorite. . .'. Page 3, Line 28. Add the word 'a' between the words 'makes' and 'transition'. Page 3, Line 29. Substitute the word 'extending' for the word 'extended'

to make the sentence correct. Page 8, Line 3. Insert the word 'is' between 'what' and 'reported'. Page 9, Line 1. 'myrmekite show' should be 'myrmekite shows'. Page 10, Line 23. This should read 'different contributions' rather than 'different contribution'. Page 13, Line 1. Remove the hyphen between 'two-rheological'. Page 13, Line 2. Place the adverb (iteratively) before the verb (applying). Page 13, Line 7. The word 'follow' should be replaced with 'follows'. Make this same correction in Lines 11 and 13 and on Page 14, Line 4. Page 14, Line 2. This should read 'is composed of' rather than 'is composed by'. Page 14, Line 4. The word proportion should be replaced with 'proportions'. Page 14, Line 4. Reorganize the last part of the sentence like this: 'are then calculated as follows, respectively'. Page 14, Line 27. Change the capital A in the word 'All' to a lower-case a. Page 15, Line 23. Delete the word 'occur'. Page 16, Line 8. The article 'the' should be placed between the words under and constant in the phrase 'Under constant strain rate assumption...' Page 16, Line 10-11. This should read: 'Strain rates on the order of...'.

Figure 1: Figure 1a: Label the minerals with the abbreviations qtz, bt, etc., to support the caption and to match Figure 1b. Figure 1c: label the myrmekite directly on the figure. Figure 1d: the caption refers to K-spar, but only plagioclase is identified here. Is this the correct figure? Figure 1f: why is the note about CL image after EBSD scan included? There is no EBSD map area labeled here, so it's hard to understand where in the figure the reader is supposed to look for this. Furthermore, is this an important and necessary point to include? Consider deleting if it's not central to the story.

Figures SOM1-5. These figures are commonly called out in the text and contain important primary data, so they need to be in the main text with the rest of the figures rather than in supplementary materials. These maps are more insightful on process than just the phase maps in the standard figures, so this is an important change to make.

---

## Author Comment (AC2) · 14 Oct 2018

Response to Anonymous Referee #1 comments (se-2018-70-RC1)

". . .In this paper, there are so many supplementary figures. These figures frequently referred in the main text, and then it is complicated and disturbs our understanding the manuscript. Some supplementary figures should be appeared as figures in the manuscript. The order of figures is somewhat strange. The results of image analysis of grain size and shape (Figs. 6 and 7) should be appeared prior to the results of phase spatial distribution analysis (Fig. 5)."

Response: We have included Figures SOM2 and SOM4 in the main text (they are the new Figures 5 and 6), as they indeed show important EBSD data. We prefer to leave

the other supplementary Figures in the SOM, as they contain data that complement and expand the figures presented in the main text. The figure order reflects the order in which figures are cited in the text. Results of image analysis are described in a paragraph after the EBSD data, therefore the figures follow the same order. The text has been changed in order to respect the order of figures: results of grain size and aspect ratio analyses (Figs. 8 and 9) are now described in a separate section following the description of EBSD data.

"Descriptions of the rheological calculations (section 6.3) are little bit complicated, and then they are not easy to understand. I would like the authors to rewrite and reorganize some sentences in the section 6.3."

Response: We have made the effort of maintaining the description of rheological calculation as simple as possible and as complete as possible. To achieve this, we have slightly modified the chapter and introduced subheadings.

"Although the authors described that micro-cataclastic process or micro-fracturing is a dominant grain size reduction mechanism of plagioclase in the samples analysed here, the microstructural observations indicative of the microcataclastic process or micro-fracturing of plagioclase are not described sufficiently."

Response: We agree that referring to microcataclastic processes is misleading and we deleted this term, as there is no evidence in the microstructure of cataclastic deformation. However, the origin of the (few) low angle boundaries within myrmekitic plagioclase (e.g. Figs. 3 and SOM 1a) requires an explanation. Given (1) the abrupt misorientation of up to 8° across such boundaries, and (2) the overall very low internal strain of the myrmekitic plagioclase, we interpret such boundaries as originating as microcracks, and not as subgrain walls resulting from recovery during crystal plastic deformation. We clarified this interpretation in the revised text, Section 6.1.2. This answers also the specific comments (10) and (19).

Specific comments: (1) P4, L2-4: In Fig. 2a, there is no identification of myrmekite and

K-feldspar for the ultramylonite. Please identify them in Fig. 2a. In Fig. 2b, there are two red bars for the ultramylonite. Is this correct? If so, what do the two different bars represent?

Response: The identification of myrmekite in the ultramylonitic layer is impossible, given that the fine-grained ultramylonitic matrix is completely mixed and no distinct layers of sheared myrmekite can be identified. K-feldspar in the ultramylonite layer occurs as rare scattered porphyroclast (as now described in the revised manuscript). In the ultramylonitic layer of Fig. 2a, no K-feldspar porphyroclasts can be detected at this magnification, but they are locally present and a good estimate is 1% of the total volume. Fig. 2b has been modified and it now contains only one single red bar. This represents the amount (area fraction) of K-feldspar porphyroclasts in the ultramylonite.

(2) P4, L16-17: What is "monocrystalline structure"? This means plagioclase is a single grain? Please clarify the structural characteristics of plagioclase within in each lobe. Response: "Monocrystalline structure" means that the plagioclase is a single grain. Monocrystalline substituted with "single grain".

(3) P4, L25-26: The quartz vermicules do not show any obvious CPO (Fig. 3d).

Response: Quartz vermicules do not show an overall CPO, but quartz vermicules WITHIN a single myrmekite lobe share a similar crystallographic orientation, as can be inferred from the clustering of few point data in the pole figures of Fig. 3d. The text has been modified accordingly.

(4) P4, L29: I do not know why "However". Please remove the word.

Response: "However" deleted.

(5) P5, L7: Please define "AR"

Response: AR: Aspect Ratio, now defined in the text.

(6) P5, L15. P9, L2: At least for me, some plagioclase grains in Area B in Fig. 4a is

elongated with the aspect ratio of >2. I would like to see the histogram for aspect ratio of plagioclase grains. Related topic also appears in P9, L2.

Response: Histograms for aspect ratio added in Figures 7e and 8d for quartz and plagioclase in sheared myrmekite, respectively.

(7) P5, L18: What does "in crystal direction" mean? It means "in the crystal coordinate system"? If so, please rephrase it.

Response: Sentence rephrased. "... misorientation axes in crystal coordinate system are almost uniformly distributed ..."

(8) P5, L19-20: two weak peaks? two strong (or distinct) peaks!

Response: "Weak" replaced with "distinct".

(9) P6, L12-15: If the pole figure of c-axis shows maxima close to Y kinematic direction, the quartz fabric pattern could be assigned to Type-II crossed gridle or single girdle with Y-point maxima. However, the authors described that the quartz fabric pattern was assigned to Type-I crossed girdle (P6, L13).

Response: Type-II crossed girdle.

(10) P8, L18–21: In this sentence, it has been described that grain size refinement of plagioclase involves micro-fracturing as suggested by misorientation analysis on the few low and high misorientation angle boundaries and CPO randomization. However, the authors have not discussed the mechanism of grain size refinement of plagioclase, based on their own microstructural observations. The following paper may be helpful to discuss this issue: Okudaira, T., Shigematsu, N., Harigane, Y. and Yoshida, K. (2017) Journal of Structural Geology, 95, 171–187.

Response: We have modified the paragraph trying to clarify the initial process of grain size refinement in plagioclase: "[. . .]Qtz grain coarsening reflects annealing of the pristine vermicular microstructure after the reaction front moved further into the Kfs (Fig.

3a), and was probably aided by dissolution-precipitation processes. Qtz coarsening implies simultaneous grain size refinement of Plg, which probably involved microfracturing, with the development of local micro-cracks in myrmekitic Plg.: Misorientation analysis on the few low and high misorientation angle boundaries inside pristine myrmekite (inside myrmekitic Plg) shows abrupt misorientations of as much as 8° across such boundaries, which could be interpreted as either micro-cracks or growth features considering the low internal distortion of grains (Figs. 3, SOM1). Microfractures could have originated from stress concentrations within the 3-D geometrically/mechanically composite structure of myrmekite (see figure 2 of Hopson and Ramseyer, 1990; Dell'Angelo and Tullis, 1996; Xiao et al., 2002). [. . .]". However the discussion here remains rather speculative, since the transition from pristine to sheared myrmekite is a dynamic process impossible to be frozen in a microstructure. The grain size refinement mechanisms during shearing of myrmekite are then discussed in the Section 6.2.1.

(11) P8,L32 – P9-L1 Kruse et al. (2001) and Miranda et al. (2016) suggested very limited deformation by dislocation creep for plagioclase aggregates in mylonites. As far as I know, Okudaira and Shigematsu (2012, Journal of Geophysical Research, 117, B03210, doi:10.1029/2011.JB008799) only described very limited deformation by dislocation creep for quartz aggregates in natural mylonites.

Response: We added a reference to the work of Okudaira and Shigematsu (2012).

(12) P9, L20: "... do not show any microstructure" may be "... do not show any deformation microstructure".

Response: Corrected.

(13) P10, L9-11: How about the effect of annealing during and or after deformation? The quartz grains associated with myrmekite may be annealed, and then some of quartz grains in monomineralic quartz layer may be also annealed at least partially.

Response: Quartz annealing probably occurs only in pristine myrmekite as a consequence of grain boundary area reduction processes and minimization of grain boundary surface energy that is expected to be high in vermicular- and in fine-grained myrmekitic quartz. On the other hand, quartz in monomineralic layers does not show microstructrues commonly considered indicative of annealing, such as 120° triple junctions, the grain shape is commonly elongate and flattened, and the grain size distribution is typically bimodal with a rather wide range. All of this suggests that quartz in the monomineralic layers was not affected by grain boundary area reduction and annealing.

(14) P11, L2: fh is water fugacity coefficient, not water fugacity itself? What is water fugacity coefficient?

Response: Yes, fh is water fugacity. "Coefficient" is redundant, deleted.

(15) P14, L2: Why would the composition of plagioclase be assumed to be An100, instead of An60? I do not understand the effect of the plagioclase composition.

Response: As reported in Rybacki and Dresen (2004), different compositions of plagioclase lead to different rheological behaviour and flow law properties. Therefore, to mimic the composition contrast between plagioclase in the granitoid rock and myrmekitic plagioclase we have adopted different plagioclase compositions in the rheological calculations for granitoid rock and myrmekite.

(16) Equations (23), (24) and (25): What does the superscript of 1 mean? Please describe them.

Response: All terms of the equations are now explained in the revised text.

(17) P14, L14: In this figure, the result of diffusion creep for quartz is not necessary.

Response: Diffusion creep of quartz in Fig. 8a. This is a deformation mechanisms map for quartz calculated with the reported flow laws for dislocation creep (Hirth et al., 2001) and thin-film pressure-solution creep (den Brok, 1998) adopted in the rheological calculation. It is important to report both deformation mechanism so one can evaluate

the conditions at which the transition occurs. Thus, we prefer to keep the diffusion creep field in the figure.

(18) P14, L18–20: I cannot understand this sentence and Fig. 8c. This sentence means a mixture of plagioclase and quartz (i.e., ideal granitoid rock) deformed by dislocation creep. The other curves in Fig. 8c are necessary? It is very confusing.

Response: Yes, they are necessary for comparison with the other curves calculated to model the rheology of different aggregates, and for the discussion developed thereafter. Quartz and plagioclase are deformed by diffusion creep? Response: In the "ideal granitoid rock", plagioclase and quartz are deforming only by dislocation creep. The calculation scheme for myrmekite is similar to those for Fig. 8b? Response: The calculation scheme for myrmekite is the same as in Fig. 8b, as now specified in the text. However in the case of the "ideal granitoid rock", the flow law is calculated only for dislocation creep.

The two technical corrections have been incorporated in the revised text.

---

## Author Comment (AC3) · 14 Oct 2018

Response to Elena A. Miranda comments (Referee #2 - se-2018-70-RC2)

1) The methods

"Regarding the methods: the methods (at least the basics) should be in the main body of the manuscript and not in the supplementary material. As written, there are not any methods in the main body. Some of the post-processing details for EBSD can stay in the supplementary material, but the instruments, working conditions, etc., need to be in the main text." Response: We have moved part of the methods (acquisition of EBSD, CL and EMPA data) to the main text, and left the post-processing details for EBSD and image analysis in the Appendix.

2) The figures

Regarding the figures: there is a tendency for important and primary EBSD data to be relegated to supplementary figures. [. . .] I recommend moving the supplementary figure EBSD maps out of the appendix, and incorporating them into the main set of figures within the manuscript." Response: See our reply to a similar comment raised by the anonymous reviewer#1.

3) Separation of data/interpretation ". . . there are a number of interpretive statements located in the geologic setting and in the data sections;"

Response: we have accepted all the suggestions by the reviewer and modified the text accordingly.

4) Flow law derivation: ". . . whereas the flow law derivation in the main body of the manuscript is extremely detailed, and much of it can be moved to the supplementary material for readers who wish to follow it in more detail."

Response: The detailed description of rheological calculation has been moved to the supplementary material. Specific Comments

(1) Abstract P1, L11: "Rephrase for clarity of reading: Here we use EBSD to investigate the microstructure of a granodiorite mylonite. . .":

Response: Sentence rephrased accordingly.

(2) P2, L11: Is it important to focus the end of the sentence on weakening of coarse-grained pegmatite? It seems like a distraction given that the focus of the paper is not on pegmatites. It may be best to remove this last phrase so that the sentence is more focused on replacement being a weakening mechanism during ductile deformation, which will provide much more continuity with the next sentence about deformation and shearing of myrmekite.

Response: The sentence has been rephrased deleting the last part and including the

citations preceding brackets.

(3) P2, L14: There needs to be a stronger link between the sentence that ends with the Viegas et al. reference and the next one that starts with 'Fine grain size'. This should be done with the second sentence (i.e., 'Fine grain size. . .) so that the weak plag + qtz aggregate is explained as a result of phase mixing and ultramylonites. As written, it does not explicitly link phase mixing and ultramylonites to the samples targeted in this study.

Response: The sentence has been rewritten in general terms, so that it doesn't need to target our specific sample. However, "the weak plag + qtz" aggregates are not the result of phase mixing in ultramylonites. What we want to show here is different: grain size reduction and phase mixing DUE TO myrmekite formation lead to ultramylonite development.

(4) P2, L16: Rephrase sentence to make it easier to read: Though the key role of myrmekite in strain localization has been recognized, it has not been accompanied with a quantitative analysis, etc. . .

Response: Sentence rephrased accordingly.

(5) P2, L18: The Introduction should be broken up into at least two paragraphs. Make a new paragraph starting with the sentence: 'Here we present the detailed analysis. . .' so that the objectives of the work are clearly separated from the background and disciplinary context.

Response: Paragraph modified accordingly.

(6) P2, L18-20: The sentence beginning with 'Here we present. . .' is awkwardly written, and needs to do a better job summarizing the findings of the work. As written, I am not sure if the authors wish to emphasize the two-fold nature of the work (e.g., 1) analysis of myrmekite evolution and 2) grain size reduction and strain localization), or if they mean to portray this as the temporal evolution of some process, i.e., starting with myrmekite

evolution and finishing with strain localization in the mylonites. The problem hinges on the part of the sentence between the words 'quartz' and 'grain size'. There either needs to be a conjunction word between 'quartz' and 'grain size' (to? and?) because it's not grammatically correct as written. This is why it is difficult to understand whether 'grain size reduction' just applies to quartz, or to both quartz and plagioclase, and calls into question the intent of the authors.

Response: The paragraph has been modified accordingly to highlight the two-fold aim of the work.

(7) P2, L21-23: These sentences are confusing as written, making it hard to understand which mylonites are being studied. If the shear zones nucleate along joints filled with quartz and epidote, where is the K-spar coming from? Are there two types of shear zones, one set of SZs within the granodiorite and the other set in the quartz- and epidote-filled joints? Does one grade laterally into the other? Please revise for clarity.

Response: The shear zones in the Rieserferner pluton nucleated on precursor joints and on quartz- and epidote- veins. As described in the referenced papers (Ceccato et al., 2017; Ceccato and Pennacchioni, 2018), homogeneous shear zones developed within quartz veins, whereas heterogeneous shear zones developed in the host rock at the immediate vicinity to joints and epidote-veins. K-feldspar is not included in the mineral assemblage of any of the veins, whereas it forms part of the original magmatic assemblage of tonalite and granodiorite of the Rieserferner pluton. The sentence has been modified to clarify these aspects.

(8) P3, L5: I recommend you remove the word 'precursor'. It is implied that the joints are precursory by saying shear zones exploited them.

Response: Deleted.

(9) P3. Section 3: Sample description and microstructure. The writing in this section is a little disjointed, with abrupt changes between sentences. Some better linkage

between sentences to provide a smoother train of thought would be helpful.

Response: The section has been splitted into two separate chapters (3. Methods and 4. Microstructures). Details of the analytical techniques have been added to the section.

(10) P3, L6-9: The sentence beginning with 'Nucleation. . .' is effectively a comparison of the field descriptions with what has been documented in the literature for other locations. Such a comparative statement is best left to the discussion rather than in a section devoted to field description.

Response: Paragraph deleted.

(11) P3, L13: I think this section is a little incomplete. There is only a statement describing the shear zones for epidote-filled joints, but the authors state in the previous section that there are also quartz-filled joints upon which shear zones nucleate and also regular joints (no filling) that serve as precursors to shear zones. I recommend that the authors add field descriptions of the shear zones associated with quartz-filled joints and "plain" joints.

Response: Paragraph has been modified accordingly.

(12) P3, L15-16: Rearrange the sentence and break into two sentences for clarity: Polished thin sections of granodiorite mylonite were prepared for the study of microstructure and of crystallographic preferred orientation (CPO). The rock chips were cut parallel to the lineation and perpendicular to the shear plane (XZ plane of finite strain ellipsoid).

Response: Paragraph modified accordingly.

(13) P3, L18: Specify which minerals were analyzed by microprobe.

Response: K-feldspar, Plagioclase. Paragraph modified accordingly.

(14) P3, L20: The description of EBSD and electron microprobe methods and analytical

conditions must be reported in the main body of the manuscript, not hidden in the Appendix.

Response: See our reply to the main comment 1).

(15) P3, L22: The sentence beginning with 'The magmatic plagioclase' is awkwardly written, too long, and hard to understand. It reads like the oscillatory zoning has a range in composition rather than the plagioclase having a range in composition.

Response: Sentence rephrased accordingly.

(16) P3, L24-28: The sentence beginning with 'Various grain size reduction...' is phrased in an interpretive rather than descriptive way. Please rephrase in terms of objective description to be consistent with the "Sample description and microstructure" section title.

Response: This is just a list of the different grain size reduction mechanisms identified looking at Rieserferner mylonite under optical microscopy. We prefer to leave this sentence, as it simply describes the observed microstructures.

(17) P4, L2: There is too abrupt of a change when the data from Figure 2 are introduced. This should be another paragraph, or at least have some more explanatory text about the volume percentage data before it is introduced. Page 4, Line 7-8.

Response: The paragraph has been separated from the preceding text.

(18) P4, L7-8: Why isn't there a photo of the ultramylonites included in Figure 1? All the rest of the mylonites described in this section have a corresponding picture, so this would be good to include for sake of completeness.

Response: As ultramylonites are not described and discussed further in the paper, we prefer to not introduce another figure.

(19) P4, L10: This section is under-developed. These sentences read like a table of contents rather than a data section. It should either be fleshed out into paragraphs

where the data are explained, or this brief summary of figure content should be merged into the text in Section 4.1 onward.

Response: Section deleted.

(20) P4, L17: Figure 7a is called out before Figures 3, 4, 5 have even been introduced. The figures should either be renumbered so that the call outs are in numerical order, or these out-of-order callouts should be removed. There is another error of this nature in Line 18 with respect to Figure 6a.

Response: Figure ordering and callout in the text have been modified.

(21) P5, L1: This sentence is strongly interpretive for a data section. It interprets the origin of the sheared myrmekite, but I suggest keeping the sample/microstructure description objective here and to wait for the Discussion to make the interpretation.

Response: We prefer to keep the terminology as it is, to underline from the beginning the strong link between plagioclase + quartz aggregates and myrmekite.

(22) P5, L9: The callout refers to figure 6e, but there is no 6e (only a-d).

Response: Corrected.

(23) P5, L6: I find the figure callouts hard to follow here. We are flipping between Figure 4 and 6, without 5 being called out yet. Perhaps rearrange the figure order to keep it more organized.

Response: Figure ordering and callout in the text have been modified.

(24) Section 4.3. The paragraphs in this section are lacking strong topic sentences to lead the paragraphs, so that the data are a bit hard to follow. Revise such that a topic sentence gives a summary of the contents of the remainder of the paragraph so that the reader has an idea of what trends are being described.

Response: Two topic sentences have been added to the beginning of each paragraph

to clarify the content of the paragraphs themselves.

(25) P5, L28: Only (100) and (010) are planes. The [001] data are directional and should be described as such.

Response: The sentence has been modified clarifying which numbers refers to planes and which to crystal directions.

(26) P7, L6: "Phase spatial distribution..." This sentence is interpretive and the importance of that interpretation is discussed in the next few sentences. I agree this is important, but the interpretation and its importance in a disciplinary context should be in the discussion rather than in the data section.

Response: The paragraph have been moved to discussions and in part deleted

(27) P10, L15-18: Are these differential stresses calculated from grain size piezometry? How are the strain rates calculated? The delivery of the stress values and strain rates is not thorough enough here; if grain size piezometry is an analysis taken on in the work, it needs to be explicitly stated.

Response: Yes, the estimates of differential stress were derived using the recrystallized grain size paleopiezometry (Stipp and Tullis 2003 and Cross et al. 2017). We have rephrased the last sentences of that chapter to clarify that those estimates were derived from the grain size distribution of the quartz domains analysed in this study.

(28) P10, L24: There is something grammatically incorrect about this sentence; revise to clarify what is meant by the last phrase.

Response: Sentence deleted.

(29) P10, Section 6.3: In the interest of brevity, it might work well to put some of the flow law derivations and calculations into the supplemental online material.

Response: The detailed description of rheological calculation has been moved to the supplementary material. However, for the sake of clarity, we have maintained a short

introduction to the rheological calculation, describing the flow laws adopted for each phase.

(30) P11, L11: It is worth noting here that this is a wet plagioclase flow law, and it would be good to quote the amount of water present in these experiments.

Response: Yes, we agree. Sentence modified accordingly.

(31) P15, L5-6: There needs to be a stronger topic sentence than the one that begins Section 6.3.1. As written, it's essentially just a callout to Figure 8, but a more powerful topic sentence would give better direction to the paragraph. The sentence should instead focus on the primary results of the deformation mechanism maps.

Response: The sentence has been modified in: "The grain-size vs. differential stress and differential stress vs. strain rate diagrams in Fig. 8 suggest the occurrence of different rheological behaviour that can be interpreted in terms of strain partitioning between aggregates with different "compositions".

(32) P15, L26: The authors state that they consider both constant stress and constant strain rate conditions, but are there any geologic data from the mylonites that support these assumptions about constant stress/strain rate? This should be justified or explained in a little more detail.

Response: Constant strain rate/stress conditions are two end-member conditions that are considered as hypothetical conditions during deformation and the consequences of such conditions are then analysed in the text.

We accepted all the technical corrections suggested, and modified the text and figures accordingly. Figure 1: Figure 1a: Label the minerals with the abbreviations qtz, bt, etc., to support the caption and to match Figure 1b. Response: Done. Figure 1c: label the myrmekite directly on the figure. Response: Done. Figure 1d: the caption refers to K-spar, but only plagioclase is identified here. Is this the correct figure? Response: Labels for Kfs have been added. Figure 1f: why is the note about CL image after EBSD

scan included? There is no EBSD map area labelled here, so it's hard to understand where in the figure the reader is supposed to look for this. Furthermore, is this an important and necessary point to include? Consider deleting if it's not central to the story. Response: The EBSD map area has now been identified by a dashed white line. It is important to highlight this fact because EBSD scans modify the CL signal for quartz and a possible reader might get confused looking at the CL image. Figures SOM1-5. These figures are commonly called out in the text and contain important primary data, so they need to be in the main text with the rest of the figures rather than in supplementary materials. These maps are more insightful on process than just the phase maps in the standard figures, so this is an important change to make. Response: See response to major comment.

---

## Author Comment (AC4) · 14 Oct 2018

Dear Editors, Dear Reviewers,

With this rebuttal letter, firstly we want to warmly thank the reviewers (Elena A. Miranda and an anonymous reviewer) for the comments and reviews that helped improve the structure of the manuscript. In addition to the changes suggested by the reviewers, major modifications include:

1. The text has been reviewed and polished to simplify sentences, disentangle complicated paragraphs and to delete repetitions or unclear statements.

2. Renumbering of the chapters; the third chapter has been split into two parts (Methods and Microstructures) to facilitate the reader and to improve the structure of the

manuscript.

3. Important data from supplementary material have been moved into the main set of figures. Data formerly included in figures SOM2 and SOM4 are now reported in figures 4 and 5.

4. Figure order and numbering have been modified accordingly.

5. The detailed description of rheological calculations applied for the elaboration of deformation mechanisms maps of Figure 9 (former Fig. 10) has been moved to the supplementary online material as suggested.

In addition to the detailed response to Reviewer's comments, attached to this rebuttal comment (Supplement) you will also find the former manuscript of the text with the tracked changes (Ceccato_etal_(Myrmekite)_TrackChanges.pdf), and the final reviewed manuscript (Ceccato_etal_(Myrmekite)_Reviewed Manuscript.pdf).

Sincerely, on behalf of all co-authors

Alberto Ceccato

Please also note the supplement to this comment:
https://www.solid-earth-discuss.net/se-2018-70/se-2018-70-AC4-supplement.zip

―――――――――――――――――

---

## Referee Report (RR1)

Dear Editor,

At your request I have reviewed the rheological aspects of the discussion in the contribution of Ceccato et al. (2018), titled 'Myrmekite and strain weakening in granitoid mylonites'.

Ceccato et al. provide a robust observational basis from which to explore the rheology of a granitoid rock that has undergone microstructural change by myrmekite formation. The authors use a combination of analytical and empirical rheological equations to make statements about the changes in strength of their samples and strain partitioning during creep.

Specifically, Ceccato et al. compare:

(1) a hypothetical granitoid rheology (which is grain size insensitive (GSI) and analytically derived);
(2) the rheology of pure quartz layers (for both grain size sensitive (GSS) and GSI creep);
(3) and the rheology of sheared myrmekite, which is calculated from microstructural observations of grain size and phase proportions (again for both GSS and GSI creep).

The authors find that the rheological analysis supports the assumption that the myrmekite, produced sny-kinematically, is weaker than the rest of the mylonite's constitutive parts. The interpreted weakness is then postulated to promote strain partitioning in the mylonite.

**General comments:**

With the inclusion of the rheological discussion the authors provide the contribution with a rounded perspective. Specifically they advance our understanding by applying the results of laboratory and theoretical works to a natural example. This links the microstructural analysis to the current understanding of the physics of creep. I particularly like the inclusion of the experimental results of Xiao et al. (2002) for comparison. This anchors the analytical rheological equation used for the sheared myrmekite to a comparable phenomenological work. It is also of note that the protolith used for comparison is not assumed to be a mono-mineralic flow law but that the authors construct a poly-mineralic law for a granitoid. I think this makes for a much better comparison and brings us closer towards the complexity of nature. The work is considered and well written and I recommend it for publication.

That being said, I have a few minor comments that may make the work clearer at points for those not expert in rheology.

Best,

James Gilgannon

**Specific comments for the Authors:**

**TEXT**

P11, L29:

*The contribution of pressure-solution creep in Qtz has been calculated following the flow law for thin-film pressure-solution of den Brok (1998):*

Previously it is written that you consider diffusion creep of quartz (P11, L17) and here you talk of pressure solution with no intermediate step in explanation. As there are many diffusion creep models that all have very similar forms it is probably helpful for the reader to have a step from the statement in line L17 to L29. Therefore I would be inclined to reformulate the sentence to something like:

'The contribution of diffusion creep in quartz is considered to come from pressure-solution creep and has been calculated using the flow law for thin-film pressure-solution of den Brok (1998):'

P12, L6-14:

*For feldspar, the flow laws of Rybacki et al. (2006) [...] Details on the derivation of the deformation mechanism maps and on the calculation of the flow laws are given in the online supplementary material.*

I think here you might want to invert the order: introduce the approach used for defining the poly-mineralic aggregates and then cite the feldspar laws. Something to the effect of:

'The flow laws for poly-mineralic aggregates (e.g. sheared myrmekite and mica-free granitoid) have been calculated following the approach of Dimanov and Dresen (2005) and Platt (2015). The method allows a poly-mineralic aggregate flow law to be constructed by considering the proportional contribution of the minerals in the aggregate. The resulting flow laws for the poly-mineralic aggregates can be derived for both a GSS and GSI rheology and are outlined in detail in the supplementary material. In our calculations only quartz and feldspar are considered as minerals of the aggregates. For quartz the flow laws used are those above (eq. 1 and 2), while for feldspar, the flow laws of Rybacki et al. (2006) have been used to calculate the contribution of dislocation and diffusion creep:

EQUATION 3

where: Af is the pre-exponential factor for feldspar (MPa-n µmm s-1); d is the grain size (µm); m is the grain-size exponent (m=3 for diffusion creep; m=0 for dislocation creep); p is the confining pressure (MPa); Vact is the activation volume (m3 mol-1). Flow law parameters are listed in Table 1. Details on the derivation of the deformation mechanism maps and on the calculation of the flow laws are given in the online supplementary material.'

This is just a rough rearrangement of what you wrote but with an additional couple of statements. I think this makes the flow of this section easier. Otherwise you introduce feldspar as a rheology after you state you will only consider quartz and poly-mineralic aggregates and not pure feldspar. In this order it makes it clearer that feldspar is used as a part of the poly-mineralic calculation.

P14, L9:

*observed for a reaction progress factor...*

Prior to this there is no mention of the reaction progress factor. I am unsure where this fits in the analysis. In the supplementary material I could access I did not see any mention of this factor. From reading the previously submitted draft, I think since your revisions you have moved the mention of this parameter to the supplementary material. I would recommend that you point to where it can be found in the supplementary material.

**EQUATIONS**

I would recommend that the strain rates for each mechanism be uniquely labelled with sub/superscripts. The reason for this is that in your rheological calculations the total strain rate is assumed to be equal to the sum of a set of strain rates from those unique mechanisms. Unique labels help make this clearer. For equation 3 that might involve breaking it into two equations: one for dislocation creep and one for diffusion creep.

**FIGURE CAPTIONS**

Figure caption 9:
I think that the caption for figure 9b needs some smoothing out.

P28, L6:

*(b) A and B marked red polygons represent the differential stress range derived from piezometric calculations on pure Qtz layers (red and black stars along respective piezometric curves).*

I think that you are referring to what you take as the iso-stress values for the red polygon in fig. 9b but it is not clear because A and B are not present in figure 9b.

**FIGURES**

Figure 2:

On the version I have there is no scale, however the inclusion of a scale might help the reader.

Figure 9:

Fig 9b)
You have a lot of information in this figure. Currently, I think that the 'black box' is hard to identify because there is also the grey polygon and all of the log-log lines. If you don't wish to break up the figure, you might consider labelling the vertical lines with the corresponding grain sizes or making the 'black box' something else, like a hatched box and making the log-log lines considerably more transparent.

Fig 9c)
Here you plot pure feldspar for An100 and An60 but do not discuss it in the text. I would remove these from the plot. In your discussion you focus on comparing pure quartz, the sheared myrmekite and the mica-free granitoid and do not discuss the role of pure feldspar.

---

## Author Response (AR2)

***Answers and responses to the comments of Topical Editor and Reviewers***

*Dear Editor,*

*Thanks for your comments. We have addressed all the outstanding points raised by the reviewers and by yourself, and this has resulted in some minor modifications to the manuscript, as requested. You can find our responses in the list here below, which starts from the comments raised by yourself.*

*We are looking forward to hearing your decision.*

*Kind regards,*

*Alberto Ceccato, Luca Menegon, Giorgio Pennacchioni, Luiz Morales*

Comments to the Author:

Dear Dr Ceccato,

We have now received all outstanding reviews, please find them attached to this message. As Dr Menegon indicated in his email a few days back, you may wonder why I requested additional reviews after the first (generally positive) responses; this is because I find that the paper as such does not yet constitute a "substantial advance in the scientific understanding of the Solid Earth", as it should for publication as a research article in this journal (https://www.solid-earth.net/about/manuscript_types.html).

The significance of myrmekite has been highlighted by Dr Menegon and others in previous work, and the effects of reaction-induced softening and transition to GSS creep in mid-crustal shear zones are well-described. As is, I find that other, more specialized journals would be better hosts for your paper.

***Response:*** *The novelty of our contribution was clearly stated in the introduction of the first version of the manuscript, "Though the key role of myrmekite in strain localization has been recognized, it has not been accompanied with a quantitative analysis of the deformation mechanisms within myrmekite-derived, fine-grained Plg + Qtz aggregates". The current paper provides an attempt to quantify the rheological effects of development of myrmekite during mylonitization based on a robust, modern microstructural analysis and on the determination of the deformation mechanisms in syn-kinematic monomyneralic and polymineralic aggregates. Such a quantitative estimate was not attempted yet in the many published studies on myrmekite in deformed rocks (e.g. LaTour and Barnett, 1987; Simpson and Wintsch, 1989; MacCaffrey, 1994; O'Hara et al., 1997; Tsurumi et al., 2003; Pennacchioni, 2005; Menegon et al., 2006; Pennacchioni and Zucchi, 2013; De Toni et al., 2016). To make the significance of our manuscript clearer in the advance of the scientific understanding of deformation of mid-crustal rocks we have now strengthened and expanded our original sentence in this last revised version.*

*The fact that our manuscript potentially represents a substantial contribution to scientific progress within the scope of Solid Earth (substantial new concepts, ideas, methods, or data) is also recognized by the 4/5 referees who all scored this point ("substantial contribution ....") as GOOD and the importance of our contribution is also highlighted in the comments of some referees:*
*Referee #1 (first revision loop)" The descriptions of microstructures and textures of feldspars and quartz presented in this paper are robust, and discussion and conclusions are reliable and interesting.*

*This paper contributes to understand deformation process/mechanism of the mid-upper continental (felsic) crust."*

*Referee #2 (Elena Miranda): "This work is an important contribution to our understanding of how quartzofeldspathic rocks deform in the middle crust where the rheology-controlling minerals of the crust (quartz and feldspar) undergo a variety of deformation mechanisms that are aided by metamorphic reaction and reaction with fluids."*

*Referee #3 (James Gilgannon): "With the inclusion of the rheological discussion the authors provide the contribution with a rounded perspective. Specifically they advance our understanding by applying the results of laboratory and theoretical works to a natural example. This links the microstructural analysis to the current understanding of the physics of creep."*

I do agree with the reviewers though that your manuscript reports some very fine science and the craftsmanship is generally good. I do also think that your manuscript can be reworked to emphasize the novelties more, and highlight how it contributes to us better understanding strain localization in the middle continental crust. By implementing such improvements, I think your paper would become suitable for publication in Solid Earth. In this context, I would invite you to rework the introduction to lead to the formulation of a clear research question or hypothesis.

**Response:** *We have modified the introduction to clarify the originality of our contribution (see our response above).*

I would also insist that you begin your data presentation with a clear contextualization of your sample(s), that includes SampleID, GPS coordinates and a geo-referenced (outcrop-scale) map that illustrates the structural, mesoscale context.

**Response:** *We have added in the Section 2 (Geological setting and field description) the requested information: "In this study we analyse a sample of mylonitic shear zone within the Rieserferner granodiorite (sample ID: 10-019A; sample coordinates: N 46°55'24.8" E 12°07'36.2"). The structural, mesoscale context of the studied sample is well illustrated by Ceccato and Pennacchioni (2018) and in particular by their Fig. 4 of their supplementary online material.*

I would then urge you to improve the structure of your discussion so as to clearly address the working hypotheses/research question, discussing your strongest arguments/most significant findings first. As is, the discussion is poorly structured and presents a lot of information in a relatively disengaging way.

**Response**: *A section header has been added to the Discussion section, to help the reader to follow the structure of the discussion as is.*

As part of this restructuring, I would recommend bringing section 7.3 and the construction of the deformation mechanism maps as the last part of your results section (these are part of your analysis). Section 7.3.1, from p13 line13 onward, should stay in the discussion. This frees up the discussion for placing your findings in the context of established knowledge, including existing knowledge of the shear zones in the Rieserferner pluton.

**Response:** *The section has been moved to the data section.*

In addition to the above, and the points raised by Referees #1,3 and 4, I would also ask you to consider the following minor comments:

P3, Section 2

Why do you know that the strain gradient you are interpreting is indeed reflecting a progressive evolution (i.e. support the assumption that space is a proxy for time here)?

*Response: There is a statement in P3-Section2 where we mention explicitly this point. However, we refer to the recent review paper of Pennacchioni and Mancktelow (2018), and references therein, to interpret that (1) joints provided the structural precursor to shear zones and (2) the fluid-rock interaction at the joint selvages pre-determined the strain gradients of the shear zones. WE believe the above cited paper provides convincing evidence for this mechanism and we "buy" this interpretation as identical structures as those described by Pennacchioni and Mancktelow (2018) have been reported in the Rieserferner pluton in the recent paper by Ceccato and Pennacchioni (2018).*

P4, lines 9-17

Not all of these descriptions can be reproduced on the basis of what shown in Fig 1. Either label the figure better or show more appropriate images. All statements here should be supported by images.

*Response: In the main text, we have listed all the microstructures recognized in the granodiorite mylonite, but not all are illustrated in the figures. Some of the microstructures are reported in the text for the sake of completeness in the microstructural description, but are not discussed later and are therefore not "relevant" to the main focus of the paper. We therefore prefer to maintain the complete list of microstructures of mylonites without adding additional images to Fig. 1. We will provide high-resolution images so that it will be possible to zoom in and see clearly the microstructural details in the figure.*

P4, lines 29-30

Consider that in the final production this figure will be too small to show this properly.

*Response: This table is supposed to be published at full-page size. In addition, we will provide high-resolution images so that it will be possible to zoom in and see clearly the microstructural details. We have always taken care of this issue in our published articles.*

P5, line 5

No such clusters shown in Fig 3D. Why is the reference to Abart et al relevant here?

*Response: Sentence "encircled clusters in Fig. 3d" has been deleted. A similar crystallographic orientation between quartz vermicular grains is better shown in Abart et al. (2014) which is therefore recalled here.*

P5, line 8

Where in these figures is this visible? Support with annotations.

*Response: An inset showing enlargement of the EBSD map has been added to the figure (Fig. 3e). Low angle boundaries are now indicated by white arrows in Fig. 3e. Figure caption modified accordingly.*

P6, line 13

Fig SOM3 has six sub-figures, please be more specific.

*Response: OK. In Fig. SOM3b,d,f. The text has been modified accordingly.*

P6, line 18

I don't see qtz layers defining a foliation in Fig. 3 and 4 and not SPO inclined to the foliation in Fig. 5.

*Response: The quartz layers clearly define the mylonitic foliation in Fig. 2. We admit this was not clear as the figure caption (that has been now corrected) did not reported that the quartz layers were coloured in black; then quartz layers "that generally define the mylonitic foliation" are partially reported in Figs. 4 and 5. The SPO inclination is in effect very weak, but this is not relevant to the topic addressed in the discussion.*

P6, line 20-22

This is a weak girdle at maybe 70 degree to XY and even lower angles to the foliation shown in Fig. 5a. The maxima of c and are 30 degrees off X and Y!

*Response: The sentence has been modified to: "…Type-II girdle inclined at high angle to the local mylonitic foliation."*

P6, line 32

See comment above, pls be specific!

*Response: Figure callout modified accordingly.*

P7, line 5, also Fig 6

This seems like a small sampling area - how did you ensure it is representative?

*Response: The area reported in Fig. 6 is just an example of one of the MANY (to be specific 5 - five) phase maps on which phase distribution analysis has been performed. We have added a sentence in the main text to make this clear: "The results obtained from the analysis of the area shown in Fig. 6 are consistent with the results obtained from other 5 areas (not presented here for the sake of brevity)".*

Section 7.2.1 start with a relevant statement (assert), then justify.

*Response: OK. We have moved (and modified) the sentence that originally concluded the 1st paragraph as an introductive assertive sentence in the new text: "The microstructures of sheared myrmekite are consistent with GSS creep, including fluid-assisted grain boundary sliding (GBS) (Boullier and Gueguen, 1975; White, 1977; Stünitz and Fitz Gerald, 1993; Fliervoet et al., 1997; Jiang et al., 2000; Wheeler et al., 2001; Lapworth et al., 2002; Bestmann and Prior, 2003; Kilian et al., 2011; Menegon et al., 2013).*

P11 line 3

"grain sizes from >40", units missing

*Response: µm added.*

line 12-13

Given the amounts of sheet silicates obvious in Figure 1, what justifies this step?

*Response: Even though the amount of sheet silicates is relevant and it could contribute significantly to the rheology of the real, natural, mylonite; in our calculations we have dealt with a simplified model in which the phases are just quartz and feldspars. In addition, flow laws and parameters for sheet silicates (biotite for example) are not well constrained as those reported for quartz and feldspars. Considering also the contribution of sheet silicates to the rheology of the rock would increase the similarity of the model to the natural case but would also increases exponentially the already existing uncertainties. Please consider the comment of Reviewer #4 that "believes" that complicated models cannot capture the complexity of nature! In contrast, we believe that our model is complicated (or if you prefer simple) enough to provide semi-quantitative information on the rheological behaviour of felsic rocks at mid-crustal conditions (opinion which is shared by Reviewer #2- Elena Miranda, and by Reviewer #3 – James Gilgannon, for example). The fit with field observations of the different extent of strain localization in felsic lithologies (see lines P12-L31 to P13-L12 in previously submitted manuscript) and with experimental data for synthetic poly-phase mixtures (e.g. Xiao et al., 2002) support the relevancy of our model.*

P12 line 9

"confining pressure"? Lithostatic pressure?

*Response: The term "confining pressure" is adopted by Rybacki et al. (2006) in the description of the flow laws for feldspars. Here we have assumed the "confining pressure" as the lithostatic pressure.*

P13 line 2

in my version of the pdf, the brackets contain 4-10-20-100 - what does this mean?

*Response: the number were supposed to represent the grain sizes of quartz for which the red curves in Fig. 9b have been calculated. Sentence deleted.*

line 3

This statement seems a bit far-fetched. Please elaborate in detail how field observations are supporting your claim.

*Response: Our statement is explained by the following sentence in the text inclusive of numerous references. All the reported references well documented that quartz veins under mid-crustal conditions sharply localized ductile shearing within granitoids. Pennacchioni (2005 and, Pennacchioni and Zucchi (2013) also did show that pegmatite layers, though coarser grained, localize strain within granitoid given the weakening effect of myrmekite.*

P14 line 3

Please use section headers to preview the section (The effect on what?)

*Response: Section header modified accordingly.*

line 11

reconsider the use of justify here - this does not seem appropriately used

*Response: "justify" changed to "could explain".*

P17 lines 10-12

This information should be given in the caption of figure 7.

*Response: Sentence moved to caption of Figs. 7 and 8.*

P20 line 13

Fix citation of De Toni et al

*Response: Done.*

Fig 7a - 54 grains - is this really statistically representative? What are the blue curves (I know you mention this in the appendix)? All information necessary to read a figure should be provided in the caption.

*Response: see comment above. I know that this amount of data is not statistically representative but this is what we have. In any case, the grain size distribution is similar to the distribution in Fig. 7b, showing the data from a statistically more significant number of grains.*

Fig 7c, d - can you please present the data behind these two histograms. In Fig. 4c, you show a value that is less than 1/96th (0.0104, <1.625 um), in 4d two values that are apparently less than 1/86th

(0.0116, <4 um). - Especially, since you cut off the analysis at >=4 um (as you say in A2). Why normalize to 100?

*Response: The cut off value in grain equivalent area is always put at 1 μm². 4 or 9 are the number of PIXELS that, according to the step-size used for EBSD mapping, represent an area of 1 μm². We are sorry but we do not fully understand this comment.*

I look forward to receiving your revised paper.

With kind regards,

Florian Fusseis

**Report #1**

I reviewed this paper previously, and recommended it to be major revision. In the revised version of the manuscript, main text and figures are modified essentially along with my comments. However, some points in the manuscript as described below should be reconsidered or revised. After sufficient modification, I recommend the revised manuscript will be accepted.

(1) P2, L5–: According to Kretz (1983), abbreviation of plagioclase is Pl, not Plg.

**Response:** *Reference to Kretz (1983) deleted. The mineral abbreviation is defined when the mineral is first cited in the text.*

(2) P4, L1: Please delete "EM".

**Response:** *done.*

(3) P5, L5: Where are "encircled clusters" in Fig. 3d.

**Response:** *This comment had also been raised by the Topical Editor (see our response above). The sentence has been deleted in the new text.*

(4) P8, L10: There is no "section 6.1.2.". It may be "section 7.2.1".

**Response:** *sentence modified accordingly.*

(5) P11, L28: The unit of the gas constant should be "J K–1 mol–1", "J/K/mol" or "J/(K·mol)".

**Response:** *corrected to J/K/mol.*

(6) P12, L12–14: At least for me, I cannot find any "details" in supplementary material.

**Response:** *"Details" refers to the complete description of calculation methods.*

(7) P13, L26–27: Which do you prefer grain size reduction of Qtz resulted from higher strain or higher stress at the margins of the monomineralic layers? If you prefer the former, grain size of Qtz in the layers is not the steady-state one. Is this OK?

*Response: Grain size reduction occurred due to an increase in strain toward the margins of monomineralic layers. This could be the result of differential strain rates between microstructural domains (such as monomineralic and polymineralic layers). This is actually a detail, described to explain the variation in grain size that we observe, and that we think is the rule, rather than the exception. Commonly this detail is omitted in the literature. Usually in polymineralic rocks, the recrystallization grain size of pure quartz layers is assumed to reflect the steady-state stress conditions. To avoid misunderstandings, the sentence has been deleted.*

(8) P14, L7–8: Please describe the stress condition for "3–4 orders of magnitude increase of strain rate".

*Response: Added: "at differential stress conditions of 70 MPa."*

(9) P14, L9: Please delete "for a reaction progress factor of □ = 0.25, i.e.", because the sentences related to the reaction progress have been omitted in the revised main text.

*Response: deleted.*

(10) P14, L30–31: Where do the values of strain rate and differential stress come from?

*Response: These values are bracketed by the curves calculated for 10vol% and 20vol% of myrmekite replacement in Fig. 9d. This information has been added in the text.*

(11) Fig. 9b: For the model myrmekite, both Pl and Qtz deformed by dislocation creep and by diffusion creep in the dislocation- and the diffusion-creep fields, respectively? If so, grain sizes of Pl and Qtz for the deformation mechanism switch are similar to each other. Perhaps, the mechanism switch of the modeled myrmekite is mainly controlled by that of plagioclase.

*Response: Yes, indeed. Probably it is controlled by that.*

By the way, the sentence of P11, L19–20 maybe not correct; grain size is not set to be 7 µm in Fig. 9b, but a variable.

*Response: Yes, we agree, this is probably a refuse of previous versions. Sentence deleted.*

(12) Fig. 9c: For the model myrmekite, grain sizes of Qtz and Pl are set to be 7 µm, and they would not change during deformation. In this case, 7 µm Pl deformed mainly by diffusion creep in the investigated conditions, whereas 7 µm Qtz deformed by diffusion creep at lower differential stresses and by dislocation creep at higher stresses. If so, please describe it in the main text.

*Response: This kind of discussion are far beyond the possibilities of the model. In the calculation of myrmekite rheology we are, in some way, averaging the properties of both phases to create a "synthetic" phase with those properties. It is therefore "impossible" to analyse what happens to the individual phases of the aggregate. Our approach is consistent with published flow laws from polymineralic aggregates, which typically derive bulk flow law parameters, rather than parameters for the individual phases (e.g. Dimanov and Dresen, JGR2005).*

**Report #3**

**Comments on the manuscript: «Myrmekite and strain weakening in…" by Ceccato, Menegon, Pennacchioni and Morales**

This is the second round of reviews after two reviewers have already commented on the manuscript. The comments of the previous reviews have been taken into account during the preparation of this revised version, and the manuscript seems to be more or less ready for publication. Every publication can be improved in one way or another, and the number of different opinions about such changes will depend on the number of reviewers chosen. In order not to extend the review process ad infinitum, I will restrict my comments to some general thoughts and to some aspects, which are incorrect in the present manuscript.

The manuscript documents a strain gradient with parallel reaction progress in a granitoid. The data appears to be of very good quality, and the conclusions are mostly sound and all based on the data presented. I have enjoyed reading the manuscript and I think the material presented is of very good quality.

General aspects:

(1) The P,T-estimates for the myrmekite reaction seem to be very low with 4kb and 420-460C. I am sure that the pseudosection modeling was carried out at the state of the art, and I did not check this in the PhD thesis or other papers. I do not know why the temperatures are so low, but they seem to be too low given some experimental data by Goldsmith, Matthews, and Spear. The problem is the following: The CaAl-silicate (zoisite, epidote) forming reaction is the reaction which separates the stability of albite + CaAl silicate from the stability field of an intermediate plagioclase. For the formation of myrmekite, you need to be on the high temperature side of this reaction, because the Al-conserving feldspar replacement reaction is the reason for the quartz precipitates in the symplectite microstructure. The CaAl-silicate reaction delineates the boundary between greenschist and amphibolite facies, at least in mafic rocks. The only way to stabilize an intermediate plagioclase at low temperatures is at very low pressures, e.g. 3 kb and lower. The 4kb given here seem to be too high for that, so that the lower temperature end of the 420-460C interval certainly is too low, because that definitely is in the greenschist facies. Furthermore, epidote-quartz veins are described in the text, so that greenschist facies conditions clearly are reached.

*Response: This comment is too vague. Temperature estimates have been conducted by pseudosection calculations and are reported in the thesis of Ceccato (2018) cited in the text. The thesis is available for evaluation under request. The results of these pseudosection calculations are on their way to be submitted as an article on a scientific journal. In this paper, written in collaboration with Dr. P. Goncalves (Uni. Franche-Comté), we will present the temperature estimates during mylonitization not only in the Rieserferner, but also for other two cases of localized shear zone formation in cooling granitoid plutons (Adamello and Sierra Nevada). In all three cases, pseudosection calculations consistently indicate that mylonitization occurred at temperature in the range between 425°C and 475°C during the pluton cooling. Pseudosection have been calculated using the bulk composition of the mylonites (obtained from XRF), which is very similar in the three cases and is considered representative of the felsic, granitoid, continental middle crust. Therefore, the above discussion about phase equilibria in "[...] at least in mafic rocks", is not applicable. Please note also that formation of myrmekite in deformed rocks has been reported to occur at P-T conditions even lower than those estimated for the Rieserferner (i.e. at 370-400°C, 3-5 kbar, Tsurumi et al., 2003). Thus, we are confident that our P-T estimates are solid and well within the known range of conditions of formation of myrmekite in deformed granitoids.*

I think that part of the problem is that the shear zones described here have a fairly extended temperature history. In figure 2, there is a well-documented strain and reaction evolution in the protomylonite and mylonite with increasing amounts of myrmekite. The ultramylonite is described as a mixture of plagioclase, epidote, biotite, etc. So, for me this looks like a first stage of the deformation history, in which the protomylonite and mylonite have formed (in the amphibolite facies, and with the myrmekite reaction), and, subsequently, the ultramylonite with a greenschist facies assemblage has formed. If biotite continues to change its composition in the amphibolite facies part of the shear zone during later lower P,T conditions, the temperature estimates may be rather low as a consequence of that, but I can only speculate on the reasons for the low temperature estimate.

The temperature estimate is critical for the rheology calculations, of course. It would be a good idea to make it clear that the temperature estimates chosen for the rheological modeling are at the minimum limit for these shear zones.

**Response:** *We disagree with this speculative interpretation. The microstructural and petrographic evolution of these rocks (Adamello and Rieserferner) has been described thoroughly in the paper of Bestmann et al., 2015 and in the PhD thesis of Ceccato (2018).*

(2) The rheological model seems to follow an approach by Dimanov and Dresen (2005) and Platt (2015). As Dimanov and Dresen express it so well in their paper: "At this point it seems impossible to suggest a suitable continuum model that captures the material behavior … offering more than a purely phenomenological or qualitative description." And they have tried to model only two phases for which they have had good experimental data. The attempt made here is a very complex model with lots of different parameters, deformation mechanisms, combinations of materials, etc. I personally think that by making models more complicated we do not learn more or come closer to explaining what happens in nature. But that is my personal opinion, and as the model in this manuscript follows some approach already presented in the literature, it is up to the authors what they want to publish.

**Response**: *Yes, the model is quite complicated, with lot of different parameters and flow laws, as models often are. The paragraph of the Dimanov and Dresen's paper continues as follow: "Existing models, however, may give some guidance as to how constitutive parameters and strength of pure end-member phases vary for small volume fractions of weak or strong inclusions dispersed in a strong or weak matrix, respectively". This is the aim of this paper, trying to quantify at least in terms of order of magnitude, the relative strength of polyphase vs. monophase aggregates.*

*The results of rheological calculations presented here are then validated by comparison with both experimental data and natural examples of competence contrasts during viscous deformation of the felsic middle crust.*

For example, the den Brok model for quartz is making rather special assumptions about island and channel structures and he makes it clear that his model produces much faster strain rates than conventional diffusion creep models. But Platt (2015) chooses this model as something that is available, without giving good reasons for using it. It would perhaps have been more instructive to present a simpler model here with only end-member rheologies.

**Response:** *We do not understand which end-member rheologies should be more appropriate here. Using end-member rheologies would be rather a simple "exercise of style" and it would not show any novelty. In addition, the authors are well aware of the different models described in den Brok*

*(1998), as also Platt (2015) was. Here we use the flow law parameters described for the thin-film model. Not the island-and-channel model.*

*From den Brok (1998) paper: "It appears that the PS rate predicted for the **thin-film model** may be more than ten orders of magnitude **lower than** PS rate for **the island-channel model**. The reason for this discrepancy is that at T = 200 °C, the predicted effective grain-boundary diffusivity (Dw) for the thin-film model, where material diffuses through a very thin and structured water film, is ~10–29 m3/s (Farver and Yund, 1991), which is more than ten orders of magnitude lower than the effective diffusivity predicted for the island-channel model, where material diffuses through relatively wide channels filled with water having the properties of a bulk fluid (Dw = 5 × 10–17 m3/s, for D = 10–9 m2/s and w = 50 nm). **Note, that at T = 400 •C the predicted PS strain rate for the island-channel is about 7 orders of magnitude higher than for the thin-film model**."*

Another aspect of the rheological discussion may be that it should be more clearly stated that the model more or less assumes something close to a Reuss lower bound, because it is assumed that the weak layers are connected. It is not clear that this assumption is also made for the granitoid case, for example.

**Response:** *Yes, this is stated clearly in the Dimanov and Dresen (2005) paper, from which the applied model is developed. Added sentence in the text (Section 7.3).*

Detailed comments:

p.2, line 6-9: Myrmekite formation is ALWAYS the result of a chemical metastability of K-feldspar, not of a stress concentration. The location, where the reaction takes place, may be controlled by the stress field (among other factors).

**Response**: *together with myrmekite formation and Kfs dismantling at high stress sites, there is also the concomitant deposition of K-feldspar at dilatant grain boundaries and around porphyroclasts in stress and strain shadows (e.g. Fig. 4a, Area D or sparse Kfs grains in the matrix of Fig.3). The concomitant precipitation of K-feldspar neoblasts in dilatant sites is consistently documented in all the studies dealing with myrmekite formation in deformed granitoids, from the inspiring work of Simpson and Wintsch (1989) onwards. This is at the core of the work on myrmekite in the last decades, and has led to the consistent conclusion that stress concentrations and strain energy are the main drivers for the myrmekite reaction in deformed rocks. The review by Vernon (1991) has highlighted these aspects very well. A metastable mineral phase is not supposed to precipitate or grow. Therefore, Kfs was a stable mineral phase during deformation of Rieserferner mylonites.*

Figure 6: The surface fraction of quartz boundaries is a wrong label on the x-axis, or the data is plotted incorrectly. The quartz surface area fraction should not be 55-60%, if quartz makes up only 18 vol% of the mixture, as described in the text. The points should be plotted mirror-symmetrically on the other side of the vertical 50% line. It then also is more consistent in that the feldspar grain boundaries plot close to their expected frequency.

**Response:** *Yes, we agree that if the data are plotted mirror-symmetrically they would plot closer to the theoretical probability curves. We would have appreciated, in a constructive review, a suggestion on to why the points "should be plotted" in one or the other way.*

*The discrepancy between area fraction of quartz (vol%) and the surface fraction of quartz boundaries reported here is probably due to the different grain sizes and shapes between quartz and plagioclase in the analysed aggregates. In fact, area fraction equals the surface fraction of boundaries only in the case in which both phases show the same grain size and shape (Heilbronner and Barrett, 2014; Sections 18.1.2-18.1.3; pp. 355-356). According to the difference in grain size or shape, the volume or surface fraction should be adopted as x-axis in the probability plot (see Section 18.4.1, pp 361-363 in Heilbronner and Barrett, 2014).*

*Here we have adopted the surface fraction of boundaries as x-axis, given that the grain size is the parameter that is readily quantifiable and shows a significant difference between quartz and plagioclase. The surface fraction of quartz boundaries was calculated as the ratio between the sum of the length of quartz grain boundaries and quartz-plagioclase phase boundaries, and the sum of the length of ALL grain (quartz and plagioclase) and phase boundaries. We have checked all the calculations and the results of our image analysis on all the five phase maps, and we confirm that the plots are correct. The phase maps are available to the interested readers who would like to use them.*

p.8, line 28: Why does the quartz coarsening imply grain size reduction in plagioclase? They could both grow.

**Response:** *Honestly, we have no idea. This is simply what we observed in the microstructure, but cannot provide a clear explanation.*

p.10, line 21: please omit "strong". There is a CPO, but not a strong one.

**Response**: *done.*

Figure 9: the largest quartz grain size in Figure 9a is plotted as 30 microns, but the measured grain size distribution gives modes of 20 for the small and 70 microns for the large fraction (Figure 7d). I did not understand why, but I might have missed something here.

**Response:** *35 μm is the divide between "small" and "large" grain sizes.*

p.14, line 16-20: this inference cannot really be made, because the myrmekite reaction is a chemical reaction, and its progress is not dependent on rheology. It is correct that the strain and the reaction progress are connected here, but it is not possible to argue for more progress of the reaction based on a weakening of the rock. In other words, the reaction correlates with strain, but it does not necessarily depend on strain rate.

**Response:** *This was not the meaning of our statements and we agree that the reaction progress does not correlate with strain rate. We intended to say that it correlates with strain and with the progressive grain size reduction of the host rock. We have modified the text to be clearer.*

p.15, line 20: The phase mixing does not impede dynamic recrystallization – the grain size sensitive processes are more efficient in terms of rheology, that is all.

*Response: We agree, the term "dynamic recrystallization" is probably misleading in this context. Dynamic recrystallization, in general is related to dislocation creep mechanisms. Dislocation creep is superseded by diffusion creep when the grain size is maintained very fine as a consequence of second-phase-pinning.*

**Reviewer #4, James Gilgannon.**

Dear Editor,

At your request I have reviewed the rheological aspects of the discussion in the contribution of Ceccato et al. (2018), titled 'Myrmekite and strain weakening in granitoid mylonites'. Ceccato et al. provide a robust observational basis from which to explore the rheology of a granitoid rock that has undergone microstructural change by myrmekite formation. The authors use a combination of analytical and empirical rheological equations to make statements about the changes in strength of their samples and strain partitioning during creep.

Specifically, Ceccato et al. compare:

(1) a hypothetical granitoid rheology (which is grain size insensitive (GSI) and analytically derived);

(2) the rheology of pure quartz layers (for both grain size sensitive (GSS) and GSI creep);

(3) and the rheology of sheared myrmekite, which is calculated from microstructural observations

of grain size and phase proportions (again for both GSS and GSI creep).

The authors find that the rheological analysis supports the assumption that the myrmekite, produced sny-kinematically, is weaker than the rest of the mylonite's constitutive parts. The interpreted weakness is then postulated to promote strain partitioning in the mylonite.

General comments:

With the inclusion of the rheological discussion the authors provide the contribution with a rounded perspective. Specifically they advance our understanding by applying the results of laboratory and theoretical works to a natural example. This links the microstructural analysis to the current understanding of the physics of creep. I particularly like the inclusion of the experimental results of Xiao et al. (2002) for comparison. This anchors the analytical rheological equation used for the sheared myrmekite to a comparable phenomenological work. It is also of note that the protolith used for comparison is not assumed to be a mono-mineralic flow law but that the authors construct a poly-mineralic law for a granitoid. I think this makes for a much better comparison and brings us closer towards the complexity of nature. The work is considered and well written and I recommend it for publication.

That being said, I have a few minor comments that may make the work clearer at points for those not expert in rheology.

Best,

James Gilgannon

Specific comments for the Authors:

TEXT

P11, L29:

The contribution of pressure-solution creep in Qtz has been calculated following the flow law for thin-film pressure-solution of den Brok (1998): Previously it is written that you consider diffusion creep of quartz (P11, L17) and here you talk of pressure solution with no intermediate step in explanation. As there are many diffusion creep models that all have very similar forms it is probably helpful for the reader to have a step from the statement in line L17 to L29. Therefore I would be inclined to reformulate the sentence to something like:

'The contribution of diffusion creep in quartz is considered to come from pressure-solution creep and has been calculated using the flow law for thin-film pressure-solution of den Brok (1998):'

**Response:** *we have reformulated the sentence as suggested.*

P12, L6-14:

For feldspar, the flow laws of Rybacki et al. (2006) […] Details on the derivation of the deformation mechanism maps and on the calculation of the flow laws are given in the online supplementary material. I think here you might want to invert the order: introduce the approach used for defining the polymineralic aggregates and then cite the feldspar laws. Something to the effect of: 'The flow laws for poly-mineralic aggregates (e.g. sheared myrmekite and mica-free granitoid) have been calculated following the approach of Dimanov and Dresen (2005) and Platt (2015). The method allows a poly-mineralic aggregate flow law to be constructed by considering the proportional contribution of the minerals in the aggregate. The resulting flow laws for the polymineralic aggregates can be derived for both a GSS and GSI rheology and are outlined in detail in the supplementary material. In our calculations only quartz and feldspar are considered as minerals of the aggregates. For quartz the flow laws used are those above (eq. 1 and 2), while for feldspar, the flow laws of Rybacki et al. (2006) have been used to calculate the contribution of dislocation and diffusion creep:

EQUATION 3

where: $A_f$ is the pre-exponential factor for feldspar (MPa-n μmm s-1); d is the grain size (μm); m is the grain-size exponent (m=3 for diffusion creep; m=0 for dislocation creep); p is the confining pressure (MPa); $V_{act}$ is the activation volume (m3 mol-1). Flow law parameters are listed in Table 1. Details on the derivation of the deformation mechanism maps and on the calculation of the flow laws are given in the online supplementary material.'This is just a rough rearrangement of what you wrote but with an additional couple of statements. I think this makes the flow of this section easier. Otherwise you introduce feldspar as a rheology after you state you will only consider quartz and poly-mineralic aggregates and not pure feldspar. In this order it makes it clearer that feldspar is used as a part of the poly-mineralic calculation.

**Response:** *We agree and the paragraph was modified accordingly.*

P14, L9:

observed for a reaction progress factor…

Prior to this there is no mention of the reaction progress factor. I am unsure where this fits in the analysis. In the supplementary material I could access I did not see any mention of this factor. From reading the previously submitted draft, I think since your revisions you have moved the mention of

this parameter to the supplementary material. I would recommend that you point to where it can be found in the supplementary material.

***Response****: sentence deleted.*

EQUATIONS

I would recommend that the strain rates for each mechanism be uniquely labelled with sub/superscripts. The reason for this is that in your rheological calculations the total strain rate is assumed to be equal to the sum of a set of strain rates from those unique mechanisms. Unique labels help make this clearer. For equation 3 that might involve breaking it into two equations: one for dislocation creep and one for diffusion creep.

***Response****: Text modified accordingly.*

FIGURE CAPTIONS

Figure caption 9:

I think that the caption for figure 9b needs some smoothing out.

P28, L6:

(b) A and B marked red polygons represent the differential stress range derived from piezometric calculations on pure Qtz layers (red and black stars along respective piezometric curves). I think that you are referring to what you take as the iso-stress values for the red polygon in fig. 9b but it is not clear because A and B are not present in figure 9b.

***Response:*** *sentence deleted.*

FIGURES

Figure 2:

On the version I have there is no scale, however the inclusion of a scale might help the reader.

***Response:*** *scale included.*

Figure 9:

Fig 9b)

You have a lot of information in this figure. Currently, I think that the 'black box' is hard to identify because there is also the grey polygon and all of the log-log lines. If you don't wish to break up the figure, you might consider labelling the vertical lines with the corresponding grain sizes or making the 'black box' something else, like a hatched box and making the log-log lines considerably more transparent.

***Response:*** *black rectangle thickened.*

Fig 9c)

Here you plot pure feldspar for An100 and An60 but do not discuss it in the text. I would remove these from the plot. In your discussion you focus on comparing pure quartz, the sheared myrmekite and the mica-free granitoid and do not discuss the role of pure feldspar.

*Response: We prefer to keep the rheological curves for the different compositions of plagioclase as a useful comparison with experimental data.*

**Myrmekite and strain weakening in granitoid mylonites**

[revised manuscript text omitted]

rheological effects of the development of myrmekite during mylonitization of granitoids, and at determining the deformation mechanisms in syn-kinematic monomyineralic and polymineralic aggregates at mid-crustal conditions. This estimate was not attempted yet (e.g. LaTour and Barnett, 1987; Simpson and Wintsch, 1989; MacCaffrey, 1994; O'Hara et al., 1997; Tsurumi et al., 2003; Pennacchioni, 2005; Menegon et al., 2006; Pennacchioni and Zucchi, 2013; De Toni et al., 2016). A validation to This analysis is validated by comparison with experimental data on deformation of poly-phase mixtures. This analysis is validated by…….

Therefore, this paper aims to represent the first case study in which the deformation mechanisms are quantitatively analysed by up to dated analytical techniques (EBSD, CL) and strain localization description is accompanied by a quantitative evaluation of the stress and strain conditions. The results of rheological calculations about lithologically-controlled rheology presented here are compared to, and their validity supported by, experimental data and natural examples of compositionally dependent ductile strain localization.

**2. Geological setting and field description**

The tonalitic-granodioritic Rieserferner pluton (Eastern Alps) (Bellieni, 1978) was emplaced at ~15 km depth (0.4 GPa; Cesare et al., 2010) into the Austroalpine nappe system at 32 Ma (Romer and Siegesmund, 2003). During post-magmatic cooling, a main set of ductile shear zones exploited shallowly ESE-dipping joints, and the joint-filling Qtz and Ep veins (set 2 of Ceccato, 2018; Ceccato and Pennacchioni, 2018). The temperature of ductile shearing has been estimated at 420-460 °C based on thermodynamic modelling (Ceccato, 2018). Ductile shearing along joints and Ep-filled joints resulted in cm-thick heterogeneous shear zones with a sigmoidal-shaped foliation in the host granodiorite (Ceccato and Pennacchioni, 2018) likely reflecting fluid-rock interaction at the vein selvages (Pennacchioni and Mancktelow, 2018). In contrast, Qtz veins filling the joints sharply localized homogeneous shearing (Ceccato et al., 2017).

TheIn this study we analyse a sample of mylonitic shear zone within the Rieserferner granodiorite (sample ID: 10-019A; sample coordinates: N 46°55'24.8" E 12°07'36.2"). of heterogeneous shear zone analysed here (10-019A) comes from the outcrops just north of the glacial lake at the base of the Hochgall-Ferner, in the Rieserferner Group (GPS coordinates: N 46°55'24.8" E 12°07'36.2"). The structural, mesoscale

[revised manuscript text omitted]